# Causal Contrastive Learning
# for Counterfactual Regression Over Time

Mouad El Bouchattaoui [*,1,2], Myriam Tami[1], Benoit Lepetit[2], and Paul-Henry Cournède[1]

[1]Paris-Saclay University, CentraleSupélec, MICS Lab, Gif-sur-Yvette, France
[2]Saint-Gobain, Paris, France

## Abstract

Estimating treatment effects over time holds significance in various domains, including precision medicine, epidemiology, economy, and marketing. This paper introduces a unique approach to counterfactual regression over time, emphasizing long-term predictions. Distinguishing itself from existing models like Causal Transformer, our approach highlights the efficacy of employing RNNs for long-term forecasting, complemented by Contrastive Predictive Coding (CPC) and Information Maximization (InfoMax). Emphasizing efficiency, we avoid the need for computationally expensive transformers. Leveraging CPC, our method captures long-term dependencies in the presence of time-varying confounders. Notably, recent models have disregarded the importance of invertible representation, compromising identification assumptions. To remedy this, we employ the InfoMax principle, maximizing a lower bound of mutual information between sequence data and its representation. Our method achieves state-of-the-art counterfactual estimation results using both synthetic and real-world data, marking the pioneering incorporation of Contrastive Predictive Encoding in causal inference.

## 1 Introduction

It's vital in real-world applications to estimate potential responses, i.e., responses under hypothetical treatment strategies. Individuals show diverse responses to the same treatment, emphasizing the need to quantify individual response trajectories. This enables personalized interventions, enhancing decision-making efficacy. In medical contexts, precise response estimation enables tailored treatments for patients [2, 70, 49]. This paper focuses on *counterfactual regression over time*, estimating responses under hypothetical treatment plans based on individual records, including past covariates, responses, and treatment sequences up to the current prediction time [64, 62]. In addressing the challenges of this time-varying setting, we tackle: (1) **Time-dependent confounding** [55]: confounders influenced by past treatment, impacting subsequent treatments and responses. (2) **Selection bias:** imbalanced covariate distributions across treatment regimes in observational data, requiring time-aware handling beyond methods in static settings [64, 67, 41]. (3) **Long-term dependencies**: enduring interdependencies among covariates, treatments, and responses, enabling long-range interactions [15, 54].

Recent advancements in neural networks, such as Recurrent Marginal Structural Networks (RMSNs) [42], Counterfactual Recurrent Networks (CRN) [7], and G-Net [39], have tackled these causal inference challenges. However, their reliance on RNNs limits their ability to capture long-term dependencies. Recent studies [48] propose integrating transformers to better represent temporal dynamics. Rather than viewing this as a limitation of RNNs, we see it as an opportunity to emphasize their strengths. We design specific architectures for counterfactual regression over large horizons, avoiding complex, hard-to-interpret models like transformers. Our approach leverages

---

[*]mouad.el-bouchattaoui@centralesupelec.fr

38th Conference on Neural Information Processing Systems (NeurIPS 2024).

the computational efficiency of RNNs, incorporating Contrastive Predictive Coding (CPC) [52, 29] for learning data history representations. This enhances model performance while maintaining efficiency, offering a compelling alternative to transformer-based approaches. Furthermore, we usually formulate identification assumptions of counterfactual responses over the original process history space (Appendix B.1). However, these assumptions may not hold in the representation space for arbitrary functions. Since identification often involves conditional independence, it applies when using an invertible representation function. Current models for time-varying settings [42, 7, 48] do not enforce representation invertibility. To address this, instead of adding complexity with a decoder, we *implicitly* push the history process to be "reconstructable" from the encoded representation by maximizing Mutual Information (MI) between representation and input, following the InfoMax principle [43], akin to Deep InfoMax [32].

## 2    Contributions

Our approach is inspired by self-supervised learning using MI objectives [32]. We aim to maximize MI between different views of the same input, introducing counterfactual regression over time through CPC to capture long-term dependencies. Additionally, we propose a tractable lower bound to the original InfoMax objectives for more efficient representations. This is challenging due to the sequential nature and high dimensionality, marking a novelty. We demonstrate the importance of regularization terms via an ablation study. Previous work leveraging contrastive learning for causal inference applies only to the static setting with no theoretical grounding [16]. To our knowledge, we frame for the first time the representation balancing problem from an information-theoretic perspective and show that the suggested adversarial game (Theorem 5.4) yields theoretically balanced representations using the Contrastive Log-ratio Upper Bound (CLUB) of MI, computed efficiently. Key innovations of our Causal CPC model include: (1) We showcase the capability of leveraging CPC to capture long-term dependencies in the process history using InfoNCE [25, 26, 52], an unexplored area in counterfactual regression over time where its integration into process history modeling is not straightforward in causality. (2) We enforce input reconstruction from representation by contrasting the representation with its input. Such quality is generally overlooked in baselines, yet it ensures that confounding information is retained, preventing biased counterfactual estimation. (3) Applying InfoMax to process history while respecting its dynamic nature is challenging. We provide a tractable lower bound to the original InfoMax problem, also bringing theoretical insights on the bound's tightness. (4) We suggest minimizing an upper bound on MI between representation and treatment to make the representation non-predictive of the treatment, using the CLUB of MI [13]. This novel information-theoretic perspective results in a theoretically balanced representation across all treatment regimes. (5) By using a simple Gated Recurrent Unit (GRU) layer [14] as the model backbone, we demonstrate that well-designed regularizations can outperform more complex models like transformers. Finally, our experiments on synthetic data (cancer simulation [24]) and semi-synthetic data based on real-world datasets (MIMIC-III [35]) show the superiority of Causal CPC at accurately estimating counterfactual responses.

## 3    Related Work

**Models for Counterfactual Regression Through Time**    Traditionally, causal inference addresses time-varying confounders using Marginal Structural Models (MSMs) [64], which rely on inverse probability of treatment weighting [62]. However, MSMs can yield high variance estimates, especially with extreme values, and are limited to pooled logistic regression, impractical for high-dimensional, dynamic data. RMSNs [41] enhance MSMs by integrating RNNs for propensity and outcome modeling. CRN [7] employs adversarial domain training with a gradient reversal layer [23] to establish a treatment-invariant representation space, reducing bias induced by time-varying confounders. Similarly, G-Net [39] combines g-computation and RNNs to predict counterfactuals in dynamic treatment regimes. Causal Transformer (CT) [48] uses transformers to estimate counterfactuals over time and handles selection bias by learning a treatment-invariant representation via Counterfactual Domain Confusion loss (CDC) [78]. These models, like ours, assume *sequential ignorability* [62], in contrast to a body of work which does not fully verify our assumptions [45, 73, 68, 60, 80, 79, 8, 28, 12, 38, 58, 69, 19, 10, 31, 34, 6, 11, 82, 22], which we discuss in detail in Appendix C.1.2. In contrast, we introduce a contrastive learning approach to capture long-term dependencies while maintaining a simple model and ensuring high computational efficiency in both training and prediction. This demonstrates that simple models with well-designed regularization terms can still achieve high

prediction quality. Additionally, previous works [62, 64, 41, 39, 48] did not consider the role of invertible representation in improving counterfactual regression. Here, we introduce an InfoMax regularization term to make our encoder easier to invert. Appendix C.1 provides a detailed overview of counterfactual regression models.

**InfoMax Principle** The InfoMax principle aims to learn a representation that maximizes MI with its input [43, 5]. Estimating MI for high-dimensional data is challenging, often addressed by maximizing a simple and tractable lower bound on MI [32, 56, 40]. Another approach involves maximizing MI between two lower-dimensional representations of different views of the same input [3, 29, 75, 77], offering a more practical solution. We adopt this strategy by dividing our process history into two views, past and future, and maximizing a tractable lower bound on MI between them. This encourages a "reconstructable" representation of the process history. To our knowledge, the only work applying an InfoMax approach to counterfactual regression, albeit in static settings, is [16]. They propose maximizing MI between an individual's representation and a global representation, aggregating information from all individuals into a single vector. However, the global representation lacks clarity and interpretability, raising uncertainties about its theoretical underpinnings in capturing confounders. Furthermore, there's a lack of theoretical analysis on how minimizing MI between individual and treatment-predictive representations could yield a treatment-invariant representation. As a novelty, we extend the InfoMax principle to longitudinal data, providing a theoretical guarantee of learning balanced representations. Appendix C.2 discusses self-supervision and MI, with all proofs in Appendix G.

## 4 Problem Formulation

**Setup** In the framework of Potential Outcomes (PO) [65], and following [63], we track a cohort of individuals (units) $i \in \{1, 2, \ldots, N\}$ over $t_{max}$ time steps. At each time $t \in \{1, 2, \ldots, t_{max}\}$, we observe the following: (1) **Discrete treatment** $W_{it} \in \mathcal{W} = \{0, 1, \ldots, K-1\}$, e.g., in medical contexts, $W_{it}$ may represent treatments like radiotherapy or chemotherapy. (2) **Outcome of interest** $Y_{it} \in \mathcal{Y} \subset \mathbb{R}$, such as tumor volume. (3) **Time-varying context** $\mathbf{X}_{it} \in \mathcal{X} \subset \mathbb{R}^{d_x}$, containing information about the individual that may influence treatment decisions and outcomes. $\mathbf{X}_{it}$ is a $d_x$-dimensional vector of confounders, such as health records or clinical measurements. (4) **Static confounders** $\mathbf{V} \in \mathcal{V} \subset \mathbb{R}^{d_v}$, such as

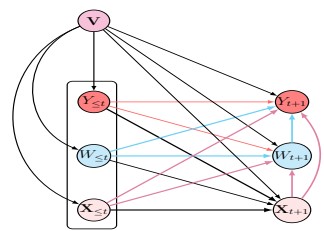

Figure 1: Causal graph over $\mathbf{H}_{t+1}$

gender, which remain constant over time. (5) **Partially observed potential outcomes** $Y_{it}(\omega_{i, \leq t})$, representing the outcomes that *would have been* observed for individual $i$ at time $t$ under treatment sequence $\omega_{i, \leq t} = (\omega_{i,1}, \ldots, \omega_{i,t}) \in \mathcal{W}^t$. We define the history process up to time $t + 1$ as $\mathbf{H}_{t+1} = [\mathbf{V}, \mathbf{X}_{\leq t+1}, W_{\leq t}, Y_{\leq t}]$, capturing all information prior to the assignment of treatment $W_{t+1}$. This history is illustrated in the causal graph shown in Figure 1.

**Goal** Given a training dataset $\{\mathbf{H}_{i, t+1}, i = 1, \ldots, N\}$ sampled from the empirical distribution $\mathbb{P}_{\mathbf{H}_{t+1}}$, we address the following causal inference problem: *Given a history process $\mathbf{H}_{t+1}$, how can we efficiently estimate counterfactual responses up to time $t + \tau$ (where $\tau \geq 1$ is the prediction horizon) for a potential treatment sequence $\omega_{t+1:t+\tau} = (\omega_{t+1}, \ldots, \omega_{t+\tau})$?* The goal is to estimate the causal quantity: $\mathbb{E}(Y_{t+\tau}(\omega_{t+1:t+\tau}) \mid \mathbf{H}_{t+1})$, i.e., the expected outcome at time $t + \tau$, given the history $\mathbf{H}_{t+1}$ and a sequence of treatments $\omega_{t+1:t+\tau}$. We identify this causal quantity from observational data using the assumption of sequential ignorability [62, 42, 39, 48], which is implicitly assumed in Figure 1 and explicitly discussed in Appendix B.1. This allows us to express the counterfactual as:

$$\mathbb{E}(Y_{t+\tau}(\omega_{t+1:t+\tau}) \mid \mathbf{H}_{t+1}) = \mathbb{E}\left(Y_{t+\tau} \mid \mathbf{H}_{t+1}, W_{t+1:t+\tau} = \omega_{t+1:t+\tau}\right).$$

## 5 Causal CPC

### 5.1 Representation Learning

**Contrastive Predictive Coding** We employ contrastive learning to efficiently represent the process history $\mathbf{H}_t$. Causal forecasting over multiple time horizons requires representations that capture variability in $\mathbf{H}_t$. For short-term predictions, local features and smooth signal variations are critical,

while long-term predictions rely on capturing global structures and long-term dependencies, as shared information between history and future points diminishes.

To achieve this, we learn a representation of $\mathbf{H}_t$ that predicts future components over multiple time steps. For each horizon $j = 1, \ldots, \tau$, future components are defined as $\mathbf{U}_{t+j} = [\mathbf{V}, \mathbf{X}_{t+j}, W_{t+j-1}, Y_{t+j-1}] \in \mathcal{U} \subset \mathbb{R}^{(d_v+d_x+K+1)}$. First, local features are extracted by encoding $[\mathbf{V}, \mathbf{X}_t, W_{t-1}, Y_{t-1}]$ into $\mathbf{Z}_t = \Phi_{\theta_1}([\mathbf{V}, \mathbf{X}_t, W_{t-1}, Y_{t-1}])$. Then, the full process history $\mathbf{H}_t$ is summarized into a *context representation* $\mathbf{C}_t$, given by an autoregressive model: $\Phi_{\theta_2}^{ar}(\mathbf{Z}_{\leq t}) = \mathbf{C}_t$, where $\Phi_{\theta_2}^{ar}$ is implemented with a GRU [14]. This results in the representation function: $\Phi_{\theta_1, \theta_2}(\mathbf{H}_t) = \mathbf{C}_t$. To train the model, we use a contrastive loss that encourages the context $\mathbf{C}_t$ to predict future local features $\mathbf{Z}_{t+1}, \ldots, \mathbf{Z}_{t+\tau}$ while distinguishing them from the features of other individuals. This is done by minimizing the InfoNCE loss $\mathcal{L}_j^{(InfoNCE)}$ for each horizon $j$:

$$\mathcal{L}_j^{(InfoNCE)}(\theta_1, \theta_2, \Gamma_j) \coloneqq -\mathbb{E}_{\mathcal{B}} \left[ \log \frac{\exp(T_j(\mathbf{U}_{t+j}, \mathbf{C}_t))}{\sum_{l=1}^{|\mathcal{B}|} \exp(T_j(\mathbf{U}_{l,t+j}, \mathbf{C}_t))} \right], \tag{1}$$

where $\mathcal{B}$ is a batch containing individual histories, and $\Gamma_j$ is a weight matrix. The discriminator $T_j(.,.)$ classifies the correct future feature among negative samples from other individuals:

$$T_j(\mathbf{U}_{t+j}, \Phi_{\theta_1, \theta_2}(\mathbf{H}_t)) = \Phi_{\theta_1}(\mathbf{U}_{t+j})^T \Gamma_j \mathbf{C}_t = \mathbf{Z}_{t+j}^T \Gamma_j \mathbf{C}_t. \tag{2}$$

In practice, $\mathbf{C}_t$ predicts $\hat{\mathbf{Z}}_{t+j}$, and prediction quality is measured by the dot product $\mathbf{Z}_{t+j}^\top \hat{\mathbf{Z}}_{t+j}$. Minimizing the InfoNCE loss $\mathcal{L}_j^{(InfoNCE)}$ provides a lower bound on the MI between the context and future features $I(\mathbf{U}_{t+j}, \mathbf{C}_t)$ [52]:

$$I(\mathbf{U}_{t+j}, \mathbf{C}_t) \geq \log(|\mathcal{B}|) - \mathcal{L}_j^{(InfoNCE)}. \tag{3}$$

For multiple forecasting horizons $j = 1, 2, \ldots, \tau$, we learn long-term dependencies by minimizing the InfoNCE loss across all horizons:

$$\mathcal{L}^{CPC}(\theta_1, \theta_2, \{\Gamma_j\}_{j=1}^\tau) \coloneqq \frac{1}{\tau} \sum_{j=1}^\tau \mathcal{L}_j^{(InfoNCE)}(\theta_1, \theta_2, \Gamma_j). \tag{4}$$

Thus, minimizing $\mathcal{L}^{CPC}$ maximizes the *shared information* between the context and future components as shown in Eq. (5), pushing the model to capture the global structure of the process over large horizons—crucial for counterfactual regression across multiple time steps:

$$\frac{1}{\tau} \sum_{j=1}^\tau I(\mathbf{U}_{t+j}, \mathbf{C}_t) \geq \log(|\mathcal{B}|) - \mathcal{L}^{CPC}. \tag{5}$$

**InfoMax Principle** We introduce a regularization term to make the context representation of the process history $\mathbf{H}_t$ "reconstructable." We leverage the InfoMax principle to maximize the MI between $\mathbf{H}_t$ and the context $\mathbf{C}_t$. However, we avoid computing the contrastive loss between $\mathbf{C}_t$ and $\mathbf{H}_t$ for two main reasons. First, $\mathbf{H}_t$ is a high-dimensional sequence, making the loss computation very demanding. Secondly, we are still interested in incorporating inductive bias toward capturing global dependencies, this time by pushing any subsequence to be predictive of any future subsequence within $\mathbf{H}_t$. Hence, we divide the process history into two non-overlapping views, $\mathbf{H}_t^h \coloneqq \mathbf{U}_{1:t_0}$, $\mathbf{H}_t^f \coloneqq \mathbf{U}_{t_0+1:t}$ representing a *historical subsequence* and a *future subsequence* within the process history $\mathbf{H}_t$, with $t_0$ randomly chosen per batch. We then maximize the MI between the representations of these views, $\mathbf{C}_t^h$ and $\mathbf{C}_t^f$, resulting in a lower bound to the InfoMax objective as formulated below:

**Proposition 5.1.** $I(\mathbf{C}_t^h, \mathbf{C}_t^f) \leq I(\mathbf{H}_t, (\mathbf{C}_t^h, \mathbf{C}_t^f))$.

We provide an intuitive discussion of the inequality by providing an exact writing of the gap in 5.1:

**Theorem 5.2.**

$$I(\mathbf{H}_t; (\mathbf{C}_t^f, \mathbf{C}_t^h)) - I(\mathbf{C}_t^f, \mathbf{C}_t^h) = I(\mathbf{H}_t; \mathbf{C}_t^f \mid \mathbf{C}_t^h) + \mathbb{E}_{\mathbf{h}_t \sim \mathbb{P}_{\mathbf{H}_t}} \mathbb{E}_{\mathbf{c}_t^f \sim \mathbb{P}_{\mathbf{C}_t^f \mid \mathbf{h}_t}} \left[ D_{KL}[\mathbb{P}_{\mathbf{C}_t^h \mid \mathbf{h}_t} || \mathbb{P}_{\mathbf{C}_t^h \mid \mathbf{c}_t^f}] \right]. \tag{6}$$

Both terms on the RHS of Eq. (6) are positive, providing an alternative proof to Proposition 5.1. When equality holds, it implies $I(\mathbf{H}_t; \mathbf{C}_t^f \mid \mathbf{C}_t^h) = 0$, indicating that $\mathbf{H}_t$ is independent of $\mathbf{C}_t^f$ given $\mathbf{C}_t^h$. This suggests that $\mathbf{C}_t^h$ retains sufficient information from $\mathbf{H}_t$ that is predictive of $\mathbf{C}_t^f$. The symmetry of MI also leads to the occurrence of the second term on the RHS when conditioning on $\mathbf{C}_t^f$. The equality in Proposition 5.1 implies $\mathbb{P}_{\mathbf{C}_t^h \mid \mathbf{h}_t} = \mathbb{P}_{\mathbf{C}_t^h \mid \mathbf{c}_t^f}$, suggesting that $\mathbf{C}_t^f$ efficiently encodes its subsequence while sharing maximum information with $\mathbf{C}_t^h$.

By considering the proposed variant of the InfoMax principle, we can compute a contrastive bound to $I(\mathbf{C}_t^h, \mathbf{C}_t^f)$ more efficiently, as the random vectors reside in a low-dimensional space thanks to the encoding. We define a contrastive loss using InfoNCE similar to Eq. (1):

$$\mathcal{L}^{(InfoMax)}(\theta_1, \theta_2, \eta) \coloneqq -\mathbb{E}_{\mathcal{B}} \left[ \log \frac{\exp(T_\eta(\mathbf{C}_t^f, \mathbf{C}_t^h))}{\sum_{l=1}^{|\mathcal{B}|} \exp(T_\eta(\mathbf{C}_{l,t}^f, \mathbf{C}_t^h))} \right]. \tag{7}$$

We use a non-linear discriminator $T_\eta$ parametrized by $\eta$ (detailed in Appendix I). The representation of the past subsequence $\mathbf{C}_t^h$ is mapped to a prediction of the future subsequence $\hat{\mathbf{C}}_t^f \coloneqq F_\eta(\mathbf{C}_t^h)$ and $T_\eta = \mathbf{C}_t^{f^\top} \hat{\mathbf{C}}_t^f$.

Theorem 5.2 and Proposition 5.1 justify using the loss in Eq. (7) by showing that our InfoMax simplification provides a valid lower bound. Thus, the contrastive loss in Eq. (7) is valid, as:

$$\log(|\mathcal{B}|) - \mathcal{L}^{(InfoMax)} \le I(\mathbf{C}_t^h, \mathbf{C}_t^f) \le I(\mathbf{H}_t, (\mathbf{C}_t^h, \mathbf{C}_t^f)). \tag{8}$$

The "mental model" behind our regularization term comes from the MI, $I(\mathbf{H}_t, (\mathbf{C}_t^h, \mathbf{C}_t^f)) = H(\mathbf{H}_t) - H(\mathbf{H}_t \mid (\mathbf{C}_t^h, \mathbf{C}_t^f))$, which can be written using entropy. Since the entropy term is constant and parameter-free, minimizing the conditional entropy $H(\mathbf{H}_t \mid (\mathbf{C}_t^h, \mathbf{C}_t^f)) \ge 0$ ensures that $\mathbf{H}_t$ is almost surely a function of $(\mathbf{C}_t^h, \mathbf{C}_t^f)$ (Appendix G.4, Proposition G.2). When MI is maximized, the theoretical existence of such a function suggests that the learned context $\mathbf{C}_t$ can decode and reconstruct $\mathbf{H}_t$.

Beyond the idea of reconstruction, it was shown that the InfoNCE objective implicitly learns to invert the data's generative model under mild assumptions [86]. Recent works [18, 44] extend this insight to multi-modal settings, which can reframe our InfoMax problem: $\mathbf{H}_t^h$ and $\mathbf{H}_t^f$ can be seen as two coupled modalities, allowing us to identify latent generative factors up to some mild indeterminacies (e.g rotations, affine mappings). We plan to extend multi-modal causal representation learning to the longitudinal setting, where we anticipate minimizing our InfoMax objective, in the limit of infinite data, will effectively invert the data generation process up to a class of indeterminacies that we conjecture to be broader and under weaker assumptions than those in current causal representation learning literature, given our focus is on causal inference rather than the identification of causal latent variables. We initiate a formal basis for this claim in Appendix G.5.

## 5.2 Balanced representation learning

**Motivation** Our goal is counterfactual regression, specifically estimating $\mathbb{E}(Y_{t+\tau}(\omega_{t+1:t+\tau}) \mid \mathbf{H}_{t+1})$. For simplicity, with $\tau = 1$, we estimate the potential outcome for a given treatment $W_{t+1} = \omega_{t+1}$, where $W_{t+1} \in \{0, 1, \ldots, K-1\}$, expressed as $\mathbb{E}(Y_{t+1}(\omega_{t+1}) \mid \mathbf{H}_{t+1})$, which under standard assumptions is identified as:

$$\mathbb{E}(Y_{t+1}(\omega_{t+1}) \mid \mathbf{H}_{t+1}) = \mathbb{E}(Y_{t+1} \mid \mathbf{H}_{t+1}, W_{t+1} = \omega_{t+1}).$$

The RHS can be estimated from data as $\mathbb{E}(Y_{t+1} \mid \mathbf{H}_{t+1}, W_{t+1}) = f(\mathbf{H}_{t+1}, W_{t+1})$. Since only one treatment is observed per individual at each time step, $W_{i,t+1} = \omega_{i,t+1}$, our model $\hat{f}$ generates counterfactual responses by switching treatments $\hat{f}(\mathbf{h}_{i,t+1}, \omega'_{t+1})$, where $\omega'_{t+1} \ne \omega_{t+1}$ (e.g., chemotherapy vs. radiotherapy). The challenge is that $\mathbf{H}_{t+1}$ and $W_{t+1}$ are not independent, introducing potential bias in counterfactual estimation [61], leading to covariate shift or selection bias. To address this, we learn a representation $\Phi(\mathbf{H}_{t+1})$ that enforces distributional balance during decoding.

**Setup** To mitigate selection bias, we leverage the context representation $\mathbf{C}_t$ of $\mathbf{H}_t$ and introduce two sub-networks: one for response prediction and one for treatment prediction, both using a mapping of the context representation:

$$\mathbf{\Phi}_t = \text{SELU}(\text{Linear}(\mathbf{C}_t)) = \Phi_{\theta_R}(\mathbf{H}_t),$$

where SELU represents the Scaled Exponential Linear Unit [37], and $\theta_R$ denotes all parameters of the representation learner, i.e., $\theta_R = [\theta_1, \theta_2]$. Following [7, 48], our objective is to learn a representation that accurately predicts outcomes while remaining *distributionally balanced* across all possible treatment choices $W_t = 0, 1, \ldots, K-1$. To achieve this, we frame the problem as an adversarial game: one network learns to predict the next treatment from the representation, while a regularization term ensures that the representation is non-predictive of the treatment.

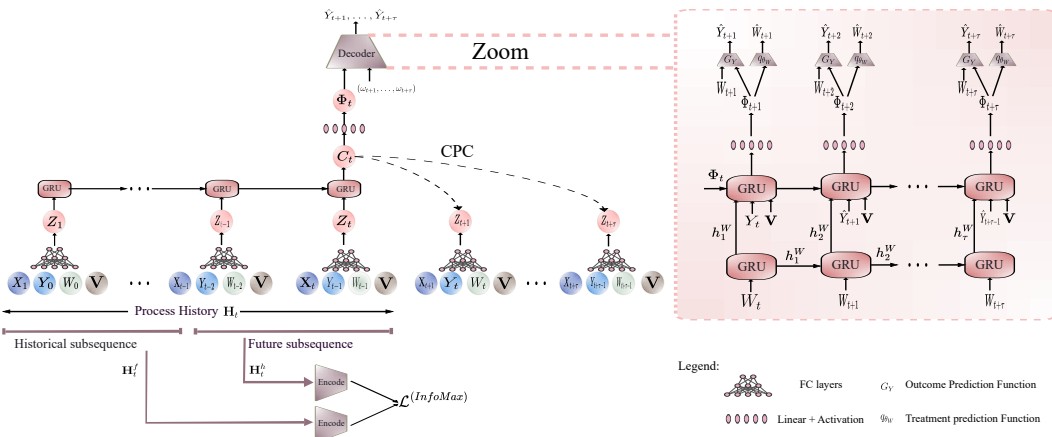

Figure 2: Causal CPC architecture: The left shows the encoder, which learns context $\mathbf{C}_t$ from process history $\mathbf{H}_t$, with CPC and InfoMax objectives used for pretraining. The right shows the decoder, which autoregressively predicts the future outcome sequence from $\mathbf{C}_t$.

**Factual response prediction**    Since we intend to predict counterfactual responses for $\tau$ steps ahead in time, we train a decoder to predict the factual responses $Y_{t+1}, \ldots, Y_{t+\tau}$ given the sequence of treatments $(W_{t+1}, \ldots, W_{t+\tau})$. We minimize the negative conditional likelihood

$$
\begin{aligned}
\mathcal{L}_Y(\theta_R, \theta_Y) &= -\log p_{\theta_Y}(y_{t+1:t+\tau} \mid \boldsymbol{\Phi}_t, \omega_{t+1:t+\tau}) \\
&= -\sum_{j=1}^{\tau} \log p_{\theta_Y}(y_{t+j} \mid y_{t+1:t+j-1}, \boldsymbol{\Phi}_t, \omega_{t+1:t+j}).
\end{aligned}
$$

We denote $\mathcal{I}_t^j := [Y_{t+1:t+j-1}, \boldsymbol{\Phi}_t, W_{t+1:t+j}]$ and assume a Gaussian distribution for the conditional responses $Y_{t+j} \mid \mathcal{I}_t^j \sim \mathcal{N}(G_Y(\mathcal{I}_t^j), \sigma^2)$, where $G_Y$ models the mean of the conditional response (see right side of Figure 2). We set $\sigma = 0.05$ throughout our experiments. The response sequence is estimated autoregressively using a GRU-based decoder without teacher forcing [81] to ensure model training's consistency with testing in real-world scenarios (Figure 2 and Algorithm 2 in Appendix H).

**Treatment prediction**    We learn a treatment prediction sub-network parameterized by $\theta_W$ that takes as input the representation $\boldsymbol{\Phi}_{t+1}$ and predicts a distribution $q_{\theta_W}(\omega_{t+1} \mid \boldsymbol{\Phi}_{t+1})$ over the treatment $W_{t+1}$ by minimizing the negative log-likelihood, $\mathcal{L}_W = -\log q_{\theta_W}(\omega_{t+1} \mid \boldsymbol{\Phi}_{t+1})$. To assess the quality of the representation in predicting the treatment, the gradient from $\mathcal{L}_W$ only updates the treatment network parameters $\theta_W$ and is not backpropagated through the response of the parameters for the representation $\boldsymbol{\Phi}_{t+1}$ (Algorithm 2, Appendix H).

**Adversarial learning**    To create an adversarial game, we update the representation learning parameters, and in the next step, the treatment network $q_{\theta_W}(\cdot \mid \boldsymbol{\Phi}_{t+1})$ with adverse losses such that the representation $\boldsymbol{\Phi}_{t+1}$ becomes invariant with respect to the assignment of $W_{t+1}$. Different from SOTA models (as highlighted in related work) and in line with our information guidelines principles, learning a balanced representation $\boldsymbol{\Phi}_{t+1}$ amounts to ensuring $\boldsymbol{\Phi}_{t+1} \perp\!\!\!\perp W_{t+1}$, which is equivalent to $I(\boldsymbol{\Phi}_{t+1}, W_{t+1}) = 0$. Hence, we minimize the MI as a way to confuse the treatment classifier. Specifically, we minimize an upper bound on $I(\boldsymbol{\Phi}_{t+1}, W_{t+1})$, namely the CLUB of MI [13].

$$
\begin{aligned}
I_{\mathrm{CLUB}}(\Phi(\mathbf{H}_t), W_{t+1}; q_{\theta_W}) := \; & \mathbb{E}_{\mathbb{P}_{(\Phi(\mathbf{H}_t), W_{t+1})}} \left[\log q_{\theta_W}(W_{t+1} \mid \Phi(\mathbf{H}_{t+1}))\right] \\
& - \mathbb{E}_{\mathbb{P}_{\Phi(\mathbf{H}_t)}} \mathbb{E}_{\mathbb{P}_{W_{t+1}}} \left[\log q_{\theta_W}(W_{t+1} \mid \Phi(\mathbf{H}_{t+1}))\right].
\end{aligned}
\tag{9}
$$

We use the objective in Eq. (9) to update the representation learner $\Phi(.)$ [9, 32]. This update aims to minimize the discrepancy between the conditional likelihood of treatments for units sampled from $\mathbb{P}_{(\mathbf{H}_t, W_{t+1})}$ and the conditional likelihood of treatments under the assumption of independent sampling from the product of marginals $\mathbb{P}_{\mathbf{H}_{t+1}} \otimes \mathbb{P}_{W_{t+1}}$. In practice, we generate samples from the product of marginals by shuffling the treatment $W_{t+1}$ across the batch dimension similar to [9, 32].

When minimizing $\mathcal{L}_W$, $q_{\theta_W}(\omega_{t+1} \mid \boldsymbol{\Phi}_{t+1})$ gets closer to the true conditional distribution $p(\omega_{t+1} \mid \boldsymbol{\Phi}_{t+1})$, and, in this case, the objective in Eq. (9) provides an upper bound of the MI between representation and treatment. We formalize the intuition by adapting the result of [13]:

**Theorem 5.3.** *[13] Let $q_{\theta_W}(\boldsymbol{\Phi}_{t+1}, \omega_{t+1}) := q_{\theta_W}(\omega_{t+1}|\boldsymbol{\Phi}_{t+1})p(\boldsymbol{\Phi}_{t+1})$ be the joint distribution induced by $q_{\theta_W}(\omega_{t+1}|\boldsymbol{\Phi}_{t+1})$ over the representation space of $\boldsymbol{\Phi}_{t+1}$. If:*

$$D_{KL}(p(\boldsymbol{\Phi}_{t+1}, \omega_{t+1})||q_{\theta_W}(\boldsymbol{\Phi}_{t+1}, \omega_{t+1})) \leq D_{KL}(p(\boldsymbol{\Phi}_{t+1})p(\omega_{t+1})||q_{\theta_W}(\boldsymbol{\Phi}_{t+1}, \omega_{t+1})),$$

*then $I(\boldsymbol{\Phi}_{t+1}, W_{t+1}) \leq I_{CLUB}(\boldsymbol{\Phi}_{t+1}, W_{t+1}; q)$.*

Based on Theorem 5.3, our adversarial training is interpretable and can be explained as follows: the treatment classifier seeks to minimize $\mathbb{E}_{\mathbb{P}_{(\mathbf{H}_t, W_{t+1})}}[\mathcal{L}_W]$, which is equivalent to minimizing Kullback-Leibler divergence $D_{KL}(p(\boldsymbol{\Phi}_{t+1}, \omega_{t+1})||q_{\theta_W}(\boldsymbol{\Phi}_{t+1}, \omega_{t+1}))$. Therefore, $q_{\theta_W}(\boldsymbol{\Phi}_{t+1}, \omega_{t+1})$ could get closer to $p(\boldsymbol{\Phi}_{t+1}, \omega_{t+1})$ than, ultimately, to $p(\boldsymbol{\Phi}_{t+1})p(\omega_{t+1})$, as we train the network to predict $W_{t+1}$ from $\boldsymbol{\Phi}_{t+1}$. In such a case and by Theorem 5.3, $I_{CLUB}$ provides an upper bound on the MI. Hence, in a subsequent step, we minimize $I_{CLUB}$ w.r.t the representation parameters, minimizing the MI $I(\boldsymbol{\Phi}_{t+1}, W_{t+1})$ and achieving balance. We theoretically formulate such behavior by proving in the following theorem that, at the Nash equilibrium of this adversarial game, the representation is exactly balanced across the different treatment regimes provided by $W_{t+1}$.

**Theorem 5.4.** *Let $t \in \{1, 2, \ldots, t_{max}\}$, $\Phi = \Phi_{\theta_R}$ and $q = q_{\theta_W}$ are, respectively, any representation and treatment network. Let $\mathbb{P}_{\Phi(\mathbf{H}_t)}$ be the probability distribution over the representation space and $\mathbb{P}_{\Phi(\mathbf{H}_t)|W_{t+1}}$ its conditional counterpart. Then, there exist $\Phi^*$ and $q^*$ such that:*

$$
\begin{aligned}
\Phi^* &= \arg\min_{\Phi} I_{CLUB}(\Phi(\mathbf{H}_t), W_{t+1}; q^*) \\
q^* &= \arg\max_{q} \mathbb{E}_{\mathbb{P}_{\Phi^*(\mathbf{H}_t)}} \left[ \log q(W_{t+1} \mid \Phi^*(\mathbf{H}_t)) \right].
\end{aligned}
\tag{10}
$$

*Such an equilibrium holds if and only if $\mathbb{P}_{\Phi(\mathbf{H}_t)|W_{t+1}=0} = \mathbb{P}_{\Phi(\mathbf{H}_t)|W_{t+1}=1} = \ldots = \mathbb{P}_{\Phi(\mathbf{H}_t)|W_{t+1}=k-1}$, almost surely.*

Since we target multi-timestep forecasting, covariate balancing in the representation space extends beyond $t+1$. For simplicity, we presented it for $t+1$, but in practice, the adversarial game applies the balancing across all forecasting horizons (Algorithm 2). The theorem also holds for other horizons by replacing $\Phi(\mathbf{H}_t)$ with $\Phi(\mathbf{H}_{t+j-1})$ and $W_{t+1}$ with $W_{t+j}$, for $2 \leq j \leq \tau$.

**Causal CPC Training**    The Causal CPC model is trained in two stages: (1) **Encoder pretraining:** We first learn an efficient representation of the process history by minimizing loss:

$$\mathcal{L}_{enc} = \mathcal{L}^{CPC}(\theta_1, \theta_2, \{\Gamma_j\}_{j=1}^{\tau}) + \mathcal{L}^{(InfoMax)}(\theta_1, \theta_2, \gamma).$$

(2) **Decoder training:** After pretraining, we fine-tune the encoder by optimizing the factual outcome and treatment networks in the adversarial game from Theorem 5.4. Formally:

$$
\begin{aligned}
\min_{\theta_R, \theta_Y} \mathcal{L}_{dec}(\theta_R, \theta_Y, \theta_W) &= \mathcal{L}_Y(\theta_R, \theta_Y) + I_{CLUB}(\Phi_{\theta_R}(\mathbf{H}_t), W_{t+1}; q_{\theta_W}), \\
\min_{\theta_W} \mathcal{L}_W(\theta_W, \theta_R) &= -\mathbb{E}_{\Phi_{\theta_R}(\mathbf{H}_t)} \left[ \log q_{\theta_W}(W_{t+1} \mid \Phi_{\theta_R}(\mathbf{H}_t)) \right].
\end{aligned}
$$

# 6    Experiments

We compare Causal CPC with SOTA baselines: MSMs [64], RMSN [41], CRN [7], G-Net [39], and CT [48]. All models are fine-tuned via a grid search over hyperparameters, including architecture and optimizers. Model selection is based on mean squared error (MSE) on factual outcomes from a validation set, and the same criterion is used for early stopping. Further details on hyperparameters and training procedures are provided in Appendices J and D.

## 6.1    Experiments with Synthetic Data

**Tumor Growth**    We use the PharmacoKinetic-PharmacoDynamic (PK-PD) model [24] to simulate responses of non-small cell lung cancer patients, following previous works [41, 7, 48]. We evaluate our approach on simulated counterfactual trajectories, varying the confounding level via the parameter $\gamma$ (Appendix E.1). Unlike [48], who used larger datasets (10,000 for training, 1,000 for testing), we use a smaller, more challenging dataset (1,000 for training, 500 for testing) to reflect real-world data limitations. For long-horizon forecasting, we set the prediction horizon to 10 and evaluate two training sequence lengths: 60 and 40, with covariates of dimension 4.

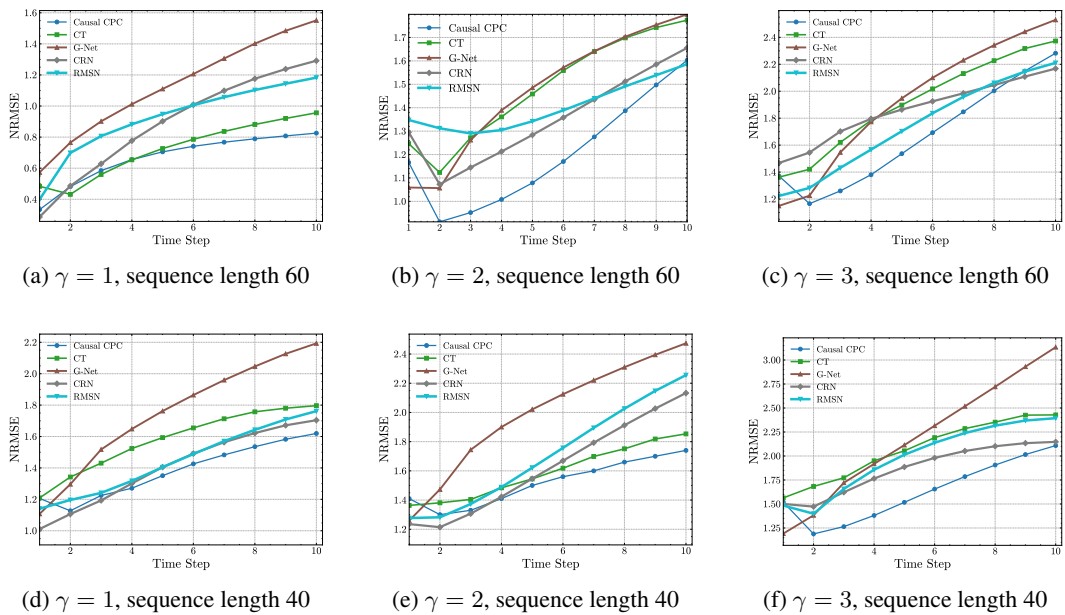

(a) $\gamma = 1$, sequence length 60    (b) $\gamma = 2$, sequence length 60    (c) $\gamma = 3$, sequence length 60

(d) $\gamma = 1$, sequence length 40    (e) $\gamma = 2$, sequence length 40    (f) $\gamma = 3$, sequence length 40

Figure 3: Evolution of error (NRMSE) in estimating counterfactual responses for cancer simulation data. Top: training sequence length 60. Bottom: training sequence length 40. In both cases, $\tau = 10$. MSM is excluded due to high prediction errors.

**Results** We tested all models on the cancer simulation data across three confounding levels, $\gamma = 1, 2, 3$. Figures 3a, 3b and 3c show the evolution of Normalized Root Mean Squared Error (NRMSE) over counterfactual tumor volume as the prediction horizon increases. Causal CPC consistently outperforms all baselines at larger horizons, demonstrating its effectiveness in long-term predictions. This confirms the quality of $\mathbf{C}_t$ in predicting future components across multiple time steps, capturing the global structure of the process as discussed in Eq. (5). Extended results are provided in Appendix E.2.1.

In the more challenging case where the maximum sequence length is 40 (Figures 3d, 3e and 3f), the error evolution remains similar to Figure 3a, 3b and 3c. Our model maintains its advantage, outperforming most baselines in long-term forecasting. However, Causal CPC does not outperform other models in short-term forecasting, a consistent limitation across experiments. Still, the model's superior long-term performance highlights its potential in applications requiring long-term accuracy. At higher levels of confounding, the model does not always outperform SOTA models at certain time steps. This may be due to the low dimensionality of the time-varying components and

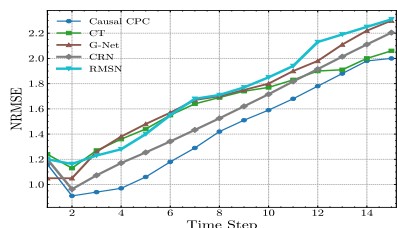

Figure 4: Models' performance for cancer simulation, $\gamma = 2$, $\tau = 15$.

static covariates, $\mathbf{U}_t = [\mathbf{V}, \mathbf{X}_t, W_{t-1}, Y_{t-1}]$, which have only four dimensions. Our model leverages contrastive learning-based regularization to excel on datasets with higher confounding dimensions, as demonstrated on MIMIC-III where $\mathbf{U}_t$ has 72 dimensions. In this setting, our model consistently outperforms baselines at longer prediction horizons. The occasional underperformance of Causal CPC at the final horizon is due to $\tau = 10$ being the last contrasted horizon, not an issue specific to $\tau = 10$. To support this, we reran all models with a sequence length of 60, $\tau = 15$, and $\gamma = 2$. As shown in Figure 4, Causal CPC still outperforms SOTA for horizons beyond $\tau = 10$ due to the encoder's retraining, where the InfoNCE loss is computed over all 15 time steps. The last prediction error remains close to SOTA, suggesting that training over larger horizons than initially intended may be beneficial.

## 6.2 Experiments with semi-synthetic and real data

**Semi-synthetic MIMIC-III** We used a semi-synthetic dataset constructed by [48] based on the MIMIC-III dataset [35], incorporating both endogenous temporal dependencies and exogenous dependencies from observational patient trajectories, as detailed in Appendix F.1. The

Table 1: Evolution of RMSEs for the semi-synthetic MIMIC III, sequence length 100.

| Model | $\tau = 1$ | $\tau = 2$ | $\tau = 3$ | $\tau = 4$ | $\tau = 5$ | $\tau = 6$ | $\tau = 7$ | $\tau = 8$ | $\tau = 9$ | $\tau = 10$ |
|---|---|---|---|---|---|---|---|---|---|---|
| **Causal CPC (ours)** | $0.32\pm0.04$ | $0.45\pm0.08$ | $0.54\pm0.06$ | $0.61\pm0.10$ | **$0.66\pm 0.10$** | **$0.69\pm0.11$** | **$0.71\pm 0.11$** | **$0.73\pm 0.06$** | **$0.75\pm 0.05$** | **$0.77\pm 0.10$** |
| **CT** | $0.42\pm 0.38$ | **$0.40\pm 0.06$** | **$0.52\pm 0.08$** | **$0.60\pm 0.005$** | $0.67\pm0.10$ | $0.72\pm0.12$ | $0.77\pm0.13$ | $0.81\pm0.14$ | $0.85\pm0.16$ | $0.88\pm0.17$ |
| **G-Net** | $0.54\pm 0.13$ | $0.72\pm0.14$ | $0.85\pm 0.16$ | $0.96\pm 0.17$ | $1.05\pm 0.18$ | $1.14\pm0.18$ | $1.24\pm 0.17$ | $1.33\pm0.16$ | $1.41\pm 0.16$ | $1.49\pm0.16$ |
| **CRN** | **$0.27\pm0.03$** | $0.45\pm0.08$ | $0.58\pm 0.09$ | $0.72\pm 0.11$ | $0.82\pm0.15$ | $0.92\pm 0.20$ | $1.00\pm 0.25$ | $1.06\pm 0.28$ | $1.12\pm 0.32$ | $1.17\pm 0.35$ |
| **RMSN** | $0.40\pm 0.16$ | $0.70\pm 0.21$ | $0.80\pm 0.19$ | $0.88\pm 0.17$ | $0.94\pm 0.16$ | $1.00\pm 0.15$ | $1.05\pm 0.14$ | $1.10\pm 0.14$ | $1.14\pm 0.13$ | $1.18\pm 0.13$ |

patient trajectories are high-dimensional and exhibit long-range dependencies. Similar to the cancer simulation, the training data consisted of relatively few sequences (500 for training, 100 for validation, and 400 for testing). Table 1 presents the mean and standard deviation of counterfactual predictions across multiple horizons ($\tau = 10$). We test two maximum sequence lengths, 100 and 60, to assess the models' robustness for long-horizon forecasting.

**Results** Causal CPC consistently outperformed the baselines, especially at larger horizons, both with a sequence length of 100 and a reduced length of 60 (Figures 1, 5). Its superior performance at longer horizons is likely due to the high number of covariates, making it well-suited to contrastive-based training. We also tested the models with 800/200/200 individuals for training/validation/testing, as in [48] (Appendix F.2.2), where Causal CPC achieved state-of-the-art (SOTA) results comparable to CT but with much shorter training and prediction times.

Figure 5: Performance for MIMIC III semi-synthetic, sequence length 60.

Table 2: Models complexity and the running time averaged over five seeds. Results are reported for tumor growth simulation ($\gamma = 1$). Hardware: GPU-1xNVIDIA Tesla M60.

| Model | Trainable parameters (k) | Training time (min) | Prediction time (min) |
|---|---|---|---|
| **Causal CPC (encoder + decoder)** | 8.2 | $16\pm 3$ | $4 \pm 1$ |
| CT | 11 | $12\pm 2$ | $30\pm 3$ |
| G-Net | 1.2 | $2 \pm 0.5$ | $35 \pm 3$ |
| CRN | 5.2 | $13\pm 2$ | $4\pm 1$ |
| RMSN | 1.6 | $22\pm 2$ | $4\pm 1$ |
| MSM | <0.1 | $1\pm0.5$ | $1\pm0.5$ |

Table 3: Ablation study with NRMSE averaged across ($1 \leq \tau \leq 10$) for cancer simulation ($\gamma = 1$) and MIMIC III.

| Model | Cancer_Sim | MIMIC III |
|---|---|---|
| Causal CPC (Full) | **1.05** | **0.62** |
| Causal CPC (w/o $\mathcal{L}^{(InfoNCE)}$) | 1.07 | 0.68 |
| Causal CPC (w/o $\mathcal{L}^{(InfoMax)}$) | 1.13 | 0.74 |
| Causal CPC (w CDC loss) | 1.07 | 0.73 |
| Causal CPC (w/o balancing) | 1.08 | 0.69 |

**Computational Efficiency and Model Complexity** Efficient execution is crucial for practical deployment, especially with periodic retraining. Beyond training, challenges arise in evaluating multiple counterfactual trajectories per individual, which grow exponentially with the forecasting horizon as $K^\tau$, where $K$ is the number of possible treatments. This is particularly relevant when generating multiple treatment plans, such as minimizing tumor volume. Table 2 shows the models' complexity (number of parameters) and running time, split between model fitting and prediction. Causal CPC is highly efficient during prediction due to its simple 1-layer GRU, similar to CRN (1-layer LSTM), while providing better ITEs estimation. In contrast, CT is less efficient due to its transformer architecture and teacher forcing, which requires recursive data loading during inference. G-Net also has longer prediction times due to Monte Carlo sampling. Overall, Causal CPC strikes a strong balance between accuracy and efficiency, making it well-suited under constrained resources.

**Real MIMIC-III Data** We evaluated our model on real MIMIC-III data, where counterfactual trajectories cannot be assessed due to the absence of observed counterfactual responses. However, performance can still be measured by forecasting factual (observed) responses over time. Our model estimates responses for each individual based on their observed treatment trajectory. As shown in Figure 6, Causal CPC consistently outperforms all baselines, especially at larger horizons, demonstrating its robustness and effectiveness in real-world settings.

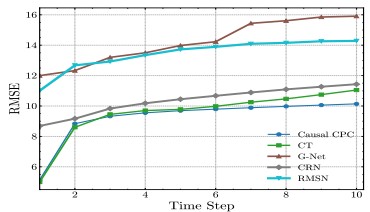

Figure 6: Evolution of RMSEs, Real MIMIC III, sequence length 100.

## 7 Discussion

**Why does Causal CPC outperform SOTA at large horizons?** Our context $\mathbf{C}_t$ is designed to capture shared information across future representations, particularly covariates, by minimizing the

InfoNCE loss over multiple time steps (Eq. 1). As shown in Eq. (5), minimizing $\mathcal{L}^{CPC}$ maximizes shared information between the context and future components, helping capture the *global structure* of the process. This is especially beneficial for counterfactual regression over long horizons, explaining the model's superior performance. However, it may not always outperform SOTA in shorter-term predictions due to its focus on long-term dependencies.

**Short-term Counterfactual Regression**   While our model is designed for long-term predictions, it may not consistently outperform SOTA for short-horizon tasks. However, the use of contrastive loss, particularly InfoNCE (Eq. 4), suggests potential adaptability to balance both short- and long-term predictions without retraining. A trade-off could be achieved by adjusting the contrastive term weights across time steps in Eq. (4), which we leave for future work.

**Ablation study**   We examined the model's performance in various configurations—full model, without CPC, and without InfoMax. Table 3 and Figure 7 show that removing either term reduces counterfactual accuracy across all horizons, underscoring their significance. Additionally, replacing our ICLUB objective with CDC loss [48] or removing balancing increases errors. We also tested different MI lower bounds like NWJ [50] and MINE [4] for both CPC and InfoMax (Appendix C.2), finding that InfoNCE yielded the best results (Table 10). Full ablation results are in Appendices E.2.2 and F.2.1.

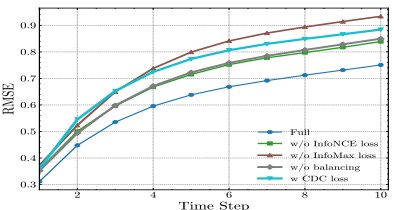

Figure 7: Ablation study of Causal CPC on MIMIC III.

**Falsifiability Test**   This study assumes sequential ignorability, common in causal inference [41, 7, 39, 48]. To assess robustness, we performed a falsifiability test by omitting certain confounders during training, while they remained in MIMIC-III data construction. As seen in Table 4, violating sequential ignorability increased prediction errors for Causal CPC, CT, and CRN, though RMSN was less affected but underperformed at $\tau \geq 2$. Despite this, Causal CPC maintained its lead at larger horizons, demonstrating strong encoding of long-term dependencies.

Table 4: Results on the MIMIC III when sequential ignorability is violated reported by RMSEs

| Model | $\tau=1$ | $\tau=2$ | $\tau=3$ | $\tau=4$ | $\tau=5$ | $\tau=6$ | $\tau=7$ | $\tau=8$ | $\tau=9$ | $\tau=10$ |
|---|---|---|---|---|---|---|---|---|---|---|
| Causal CPC | 0.44± 0.04 | 0.56± 0.07 | 0.66±0.07 | 0.73 ± 0.08 | 0.78± 0.08 | 0.83±0.06 | **0.86± 0.10** | **0.88± 0.08** | **0.91 ± 0.08** | **0.95± 0.07** |
| Causal Transformer | **0.34±0.07** | **0.48±0.07** | **0.60±0.07** | **0.68±0.06** | **0.75± 0.06** | **0.80±0.07** | **0.86± 0.09** | 0.91±0.11 | 0.95 ±0.13 | 1.00 ±0.15 |
| CRN | 0.40± 0.07 | 0.54± 0.09 | 0.70± 0.09 | 0.84± 0.09 | 0.97± 0.09 | 1.08± 0.13 | 1.18± 0.16 | 1.26± 0.19 | 1.33± 0.21 | 1.39± 0.23 |
| RMSN | 0.38± 0.08 | 0.67± 0.21 | 0.78± 0.16 | 0.84± 0.14 | 0.91± 0.14 | 0.98± 0.15 | 1.04± 0.16 | 1.09± 0.18 | 1.15± 0.19 | 1.20± 0.23 |

**Tightness of MI Upper Bounds**   Estimating MI bounds for high-dimensional variables is challenging and expensive [59, 57], often limited to low-dimensional inputs or Gaussian assumptions. In MI-constrained models, batch size is crucial. As shown in Eq. (3), increasing the batch size $\mathcal{B}$ tightens the lower bound via $\log(|\mathcal{B}|)$, and to balance memory and performance, we chose batch sizes of 256 for the encoder and 128 for the decoder. While these bounds may not be perfectly tight, mutual information and self-supervision biases significantly enhance performance (Appendix C.2), as confirmed by ablation studies. Other MI estimators like NWJ and MINE [50, 4] did not improve performance; our initial setup consistently performed better (Appendix F.2.1).

**Extending Causal CPC to Continuous Treatment**   Our approach could be extended to continuous treatments by replacing the treatment classifier with a regressor. Since the method maximizes likelihood, the equilibrium in Theorem 5.4 remains valid. However, in practice, continuous treatments will be represented by a single dimension, unlike discrete treatments with $K$-dimensional one-hot encoding. This risks losing important treatment information in counterfactual predictions. A simpler adaptation to our model could involve discretizing continuous treatments.

**Conclusion**   We introduced a novel approach to long-term counterfactual regression, combining RNNs with CPC to achieve SOTA results without relying on complex transformer models. Prioritizing computational efficiency, we incorporated contrastive loss-based regularization guided by mutual information (MI). Our method consistently outperforms existing models on both synthetic and real-world datasets, marking the first application of CPC in causal inference. Future work could focus on improving interpretability by integrating Shapley values into the causal framework [30]. Additionally, developing uncertainty-aware models tailored for longitudinal data is crucial for enhancing the reliability of predictions in our causal framework [19, 33, 84].

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

# A   Impact Statements

Our paper seeks to advance the field of Trustworthy Machine Learning by focusing on the accurate estimation of counterfactual trajectories. This capability holds significant potential to enhance decision-making processes across various domains, particularly in healthcare, where clinicians can leverage models designed to mitigate bias and promote fairness. Additionally, by focusing on efficiency, our contributions extend beyond traditional machine learning considerations to address environmental concerns associated with energy consumption. By advocating for the prudent use of computational resources, especially in training complex models deployed in real-world scenarios, we aim to promote sustainability in developing and applying machine learning solutions.

# B   Causal assumptions

## B.1   Identifiability Assumptions in Causal CPC

In this section, we detail the assumptions used for the identifiability of the counterfactual responses $\mathbb{E}(Y_{t+\tau}(\omega_{t+1:t+\tau}) \mid \mathbf{H}_{t+1})$. As briefly stated in Section 3, we follows similar assumptions to [62, 64, 7, 48], namely

**Assumption B.1** (Consistency). For every time step $t$ and given any manner by which a unit $i$ receives the sequence of treatment $\omega_{i,\leq t}$, we always observe the potential outcome $Y_{it}(\omega_{i,\leq t})$. Formally:

$$W_{i,\leq t} = w_{i,\leq t} \implies Y_{it} = Y_{it}(w_{i,\leq t}).$$

**Assumption B.2** (Sequential Ignorability). Given any time step t, we have the conditional independence:

$$Y_{it}(\omega_{it}) \perp\!\!\!\perp W_{it} | \mathbf{H}_{it} = \mathbf{h}_{it} \quad \forall(\omega_{it}, \mathbf{h}_{it})$$

**Assumption B.3** (Overlap/positivity). Given any time step $t$, and for any possible historical context $\mathbf{h}_t$, the probability of observing any of the possible treatment regimes is strictly positive but not deterministic:

$$p(\mathbf{h}_t) \neq 0 \implies 0 < p(W_t = \omega_t | \mathbf{h}_t) < 1$$

The three assumptions are sufficient for the identification of the counterfactual responses from observational data, which we formulate in the following proposition.

**Proposition B.4.** *Assuming consistency, overlap, and ignorability (assumptions B.1, B.2, B.3), the causal quantity $\mathbb{E}(Y_{t+\tau}(\omega_{t+1:t+\tau}) \mid \mathbf{H}_{t+1})$ is identifiable from observational data following*

$$\mathbb{E}(Y_{t+\tau}(\omega_{t+1:t+\tau}) \mid \mathbf{H}_{t+1}) = \mathbb{E}\left(Y_{t+\tau} \mid \mathbf{H}_{t+1}, W_{t+1:t+\tau} = \omega_{t+1:t+\tau}\right)$$

*Proof.* See [62]  □

## B.2   On the Causal Graph

We repeat the causal graph introduced in Figure 1 to explain the data generation process. here, all of the past observed data encompassed in $\mathbf{H}_{t+1}$ confounds future treatments and responses, $W_{t+1}, W_{t+2}, \ldots, W_{t_{max}}$ and $Y_{t+1}, Y_{t+2}, \ldots, Y_{t_{max}}$, which create long-term dependencies. The fact that post-covariates are affected by past treatments creates time-dependent confounding. The static covariates are assumed to be affecting all of the time-varying variables. Since we suppose sequential ignorability, there are no possible exogenous noises affecting both treatments and responses. However, such noise may possibly affect responses, time-varying covariates, and response variables.

In the figure, for simplicity, we represent past treatments as $W_{\leq t}$ such that each element in that sub-sequence confounds the next treatment and response $W_{t+1}$ and $Y_{t+1}$. Idem for $Y_{\leq t}$ and $\mathbf{X}_{\leq t}$. The static covariates $\mathbf{V}$ are assumed to be affecting all the time-varying variables. We omit the representation of exogenous noise for simplicity. Interactions between $W_{\leq t}$, $\mathbf{X}_{\leq t}$, and $Y_{\leq t}$ were also omitted for simplicity.

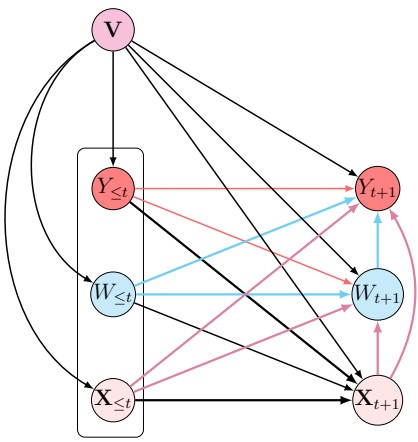

## C   Extended related work

### C.1   Counterfactual regression over time: Methods overview

#### C.1.1   Methods included in experiments

In this section, we give a brief overview of models included in our experiments: MSMs [62], RMSN [41], CRN [7], G-Net [39], and CT [48]. To delineate the differences between these models and Causal CPC, we detail in Table 5 the main design differences between all these models.

Table 5: A summary of the methods included in our experiments

| Model | Model Backbone | Tailored to long-term forecast? | Learning of long-term dependencies | Use of contrastive learning | Prediction of counter-factuals | handling selection bias | Invertibility of representation |
|---|---|---|---|---|---|---|---|
| **Causal CPC (ours)** | GRU | yes | CPC | learn long-term relations | Autoregressive | Balanced representation | yes, contrast representation with input |
| CT | 3 Transformers | yes | Transformer architecture | N/A | Autoregressive | Balanced representation | N/A |
| G-Net | LSTM | No | N/A | N/A | Autoregressive | G-Computation | Current covariates $\mathbf{X}_t$ |
| CRN | LSTM | No | N/A | N/A | Autoregressive | Balanced representation | N/A |
| RMSN | LSTM | No | N/A | N/A | Autoregressive | Weighting | N/A |
| MSM | Logistic+linear model | No | N/A | N/A | Autoregressive | Weighting | N/A |

#### C.1.2   Methods Violating Our Assumptions

Our work relies on the assumption of sequential ignorability, yet several alternative models operate under different assumptions, often addressing the presence of unobserved confounders. Some of these models are rooted in deconfounding theory [45, 60, 80], which has been extended to time-varying settings. Deconfounding involves imposing a factor model on treatment assignment, where each treatment becomes conditionally independent given latent variables that act as proxies for unobserved confounders. Examples of this approach include [8, 28, 10]. Other models assume the presence of proxy variables, inferring a representation of unobserved confounders through probabilistic models based on these proxies [79, 12, 38].

In contrast to our setting, which is governed by the three causal assumptions in Appendix B.1, many models assume a data-generating process similar to [73, 72, 58]. These methods, often non- or semi-parametric, tend to either ignore static covariates or handle them linearly, leading to computational inefficiencies and scalability issues. Nevertheless, some non- or semi-parametric approaches—such as [68, 69, 19, 31]—align with our causal assumptions but extend them to continuous time, treating sequential ignorability in a continuous setting.

Additionally, models like [34] incorporate continuous-time and assume interactions between units, where an individual's outcome depends on both their treatment and the treatments of others. [6], focusing on binary treatment sequences, requires a stronger version of sequential ignorability—conditional on current covariates—whereas our model assumes a weaker version, conditioning on the entire history of covariates to account for long-lasting confounding effects.

Furthermore, [11] focuses solely on binary treatments and targets the estimation of the average treatment effect on the treated (ATT). The authors assume a specific treatment regime, where individuals enter a post-treatment state after a defined point in time. This assumption is restrictive compared to our framework, which allows for complex, individualized treatment assignment mechanisms

and non-binary treatments, where treatment values can fluctuate over time. As a result, [11] is incompatible with our causal assumptions.

Other methods, like [82], address high-dimensional counterfactual generation based on time-varying treatment plans under the same sequential ignorability assumption. However, they are not designed for causal forecasting over multiple time steps, as required in our setting. Similarly, [22] focuses on estimating the average causal effect and is not suited for predicting individual treatment effects or conditional counterfactual responses, as it targets marginal counterfactual expectations via g-computation.

## C.2 Mutual Information and Self-Supervision

**Self-Supervised Learning and Mutual Information** In self-supervised learning, Deep InfoMax [32] uses MI computation between input images and their representations, focusing on maximizing MI to improve reconstruction quality. Local MI between representations and image patches captures detailed patterns across regions, enhancing encoding. By maximizing average MI with local regions, Deep InfoMax significantly boosts downstream task performance, while global MI plays a key role in reconstructing the entire input from the representation.

CPC aligns with the MI-based approach seen in Deep InfoMax, emphasizing the maximization of MI between global and local representation pairs. Distinct from Deep InfoMax, CPC processes local features sequentially, constructing partial "summary features" to predict specific local features in the future. While classical self-supervised paradigms often focus on tasks like classification or reconstruction-based objectives—favoring either local or global MI maximization—integrating both approaches becomes essential for downstream tasks such as counterfactual regression over time, which justifies why Causal CPC is designed to support both local and global MI maximization to improve temporal predictions.

Several other methods share similarities with CPC, such as Contrastive Multiview Coding [75]. This method emphasizes maximizing mutual information between representations of different views of the same observation. Augmented Multiscale Deep InfoMax [3], akin to CPC, makes predictions across space but differs by predicting representations across layers in the model. While Instance Discrimination [85] encourages representations capable of discriminating between individual examples in the dataset, our preference for CPC arises from its adaptability in processing sequential features in an ordered and autoregressive manner, which aligns with the requirements of our specific context, especially when dealing with counterfactual regression over time.

**Mutual Information and Inductive Bias.** Mutual information (MI) estimation success relies not only on MI's properties but also on the inductive biases from feature representation choices and MI estimator parameterization [77]. Experimental evidence shows that, although MI remains invariant under homeomorphisms, maximization with an invertible encoder during random initialization enhances downstream performance. While higher-capacity critics yield tighter MI bounds, findings consistent with [59] suggest that simpler critics provide better representations, even with looser MI bounds. Accordingly, we selected a simple bilinear critic function for contrastive losses. In vision tasks, augmentations and contrastive loss properties are crucial for representation efficiency [1, 76, 27], and [66] highlights that inductive bias, via function class representation and optimizers, significantly affects downstream performance, offering theoretical, non-vacuous guarantees on representation quality.

**Variational Approaches and MI Estimation Challenges** The estimation of MI faces inherent challenges, particularly within variational lower bounds. These bounds often degrade as MI increases, creating a delicate trade-off between high bias and high variance. To address this, methods that utilize upper bounds on MI have been developed, attempting to mitigate challenges associated with variational bounds. One strategy for MI maximization involves computing gradients of a lower MI bound concerning the parameters of a stochastic encoder. This computational approach potentially eliminates the need for direct MI estimation, providing a more tractable solution. However, estimating MI from samples remains challenging, and traditional approaches encounter scalability issues in modern machine-learning problems.

It's crucial to note that higher estimated MI between observations and learned representations does not consistently translate to improved predictive performance in downstream supervised learning tasks. CPC is an example, exhibiting less variance but more bias, with estimates capped at $\log |\mathcal{B}|$. Strategies to reduce bias, such as increasing the batch size, introduce higher computational complexity, requiring additional evaluations for estimating each batch with the encoding function.

In our empirical approach, we adopt a specific sampling strategy for sequences, considering a one-time step per batch. This facilitates computing the InfoNCE between local summary features at time t and the future prediction of local features, leading to a reduction in algorithmic complexity for contrastive loss computation. Empirical observations demonstrate non-decreased representation quality and improved prediction of factual and counterfactual outcomes.

**Other MI lower bounds.** The Mutual Information Neural Estimator (MINE) [4] leverages the relationship between MI and the Kullback-Leibler (KL) divergence. MI can be expressed as the KL divergence between the joint distribution and the product of marginals:

$$I(X; Z) := D_{KL}(\mathbb{P}_{(X,Z)} \| \mathbb{P}_X \otimes \mathbb{P}_Z)$$

MINE employs the Donsker-Varadhan representation [20] of the KL divergence:

$$D_{KL}(\mathbb{P} \| \mathbb{Q}) = \sup_{T:\Omega\to\mathbb{R}} \mathbb{E}_{\mathbb{P}}[T] - \log\left(\mathbb{E}_{\mathbb{Q}}[e^T]\right) \tag{11}$$

Here, the supremum is taken over all functions $T$ where the expectations exist. For a specific class of functions $\mathcal{F}$, potentially represented by a class of neural networks, we obtain the lower bound:

$$D_{KL}(\mathbb{P} \| \mathbb{Q}) \geq \sup_{T\in\mathcal{F}} \mathbb{E}_{\mathbb{P}}[T] - \log\left(\mathbb{E}_{\mathbb{Q}}[e^T]\right) \tag{12}$$

In practice, we maximize

$$\hat{I}_\gamma^{\text{MINE}}(\mathbb{P} \| \mathbb{Q}) = \mathbb{E}_{\mathbb{P}}[T_\gamma] - \log\left(\mathbb{E}_{\mathbb{Q}}[e^{T_\gamma}]\right),$$

where $T_\gamma$ is a discriminator parameterized by $\gamma$, representing neural network parameters. The MINE estimator is a strongly consistent estimator of the true MI (Theorem 2, [4]).

Alternatively, the **f-divergence** representation of $D_{KL}$ [51] allows us to derive another MI lower bound, known as the Nguyen, Wainwright, and Jordan (NWJ) estimator [50]:

$$D_{KL}(\mathbb{P} \| \mathbb{Q}) \geq \sup_{T\in\mathcal{F}} \mathbb{E}_{\mathbb{P}}[T] - \log\left(\mathbb{E}_{\mathbb{Q}}[e^{T-1}]\right) \tag{13}$$

This results in the estimator:

$$\hat{I}_\gamma^{\text{NWJ}}(\mathbb{P}, \mathbb{Q}) = \mathbb{E}_{\mathbb{P}}[T_\gamma] - \log\left(\mathbb{E}_{\mathbb{Q}}[e^{T_\gamma-1}]\right).$$

Unlike the InfoNCE estimator, which exhibits high bias and low variance, the NWJ estimator has a low bias but high variance [57].

# D   Experimental protocol

All models were implemented using PyTorch [53] and PyTorch Lightning [21]. In contrast to the approach in [48], we employed early stopping for all models. The stopping criterion was defined as the Mean Squared Error over factual outcomes for a dedicated validation dataset. Specifically, for the Causal CPC encoder, the stopping criterion was determined by the validation loss of the encoder.

While all models in the benchmark were trained using the Adam optimizer [36], we opted for training Causal CPC (encoder plus decoder without the treatment subnetwork) with AdamW [46] due to its observed stability during training. Similar to the common practice in training GAN discriminators, the treatment subnetwork was optimized using SGD with momentum [74].

The CT employed the Exponential Moving Average (EMA) [83] of parameters to enhance training stability. However, this technique was not applied to Causal CPC, as experimental evidence suggested only marginal improvements. Weight decay was set to zero for all models.

For each experiment, the models were trained over five different seeds, and the reported performance metrics include the mean and standard deviation of the results.

The counterfactual trajectories are generated following two strategies:

- **Single sliding treatment** [7, 48]: Trajectories are generated with a single treatment per trajectory while the treatment slides over the forecasting range to generate multiple trajectories. Similar to [7], we apply such a generation scheme to cancer simulation data.
- **Random trajectories**: Trajectories are generated such that at each time step, treatment is generated randomly. We apply random trajectories to semi-synthetic MIMIC data.

For the falsifiability test on MIMIC III datset, we mask two confounders from the inputs of the benchmark models, namely sodium and glucose measurements.

# E  Experiments on synthetic data: Details

## E.1  Description of the Simulation Model

We present a tumor growth simulation model, focusing on the PharmacoKinetic-PharmacoDynamic (PK-PD) model as discussed in [24], a recent approach to predicting treatment responses in non-small cell lung cancer patients. This simulation models the evolution of tumor volume, denoted by $V(t)$, in discrete time, where $t$ represents the number of days since diagnosis:

$$V(t) = \left(1 + \underbrace{\Lambda \log\left(\frac{K}{V(t-1)}\right)}_{\text{Tumor Growth}} - \underbrace{\kappa_c C(t)}_{\text{Chemotherapy}} - \underbrace{(\kappa_{rd} Rd(t) + \upsilon Rd(t)^2)}_{\text{Radiation}} + \underbrace{e_t}_{\text{Noise}}\right) V(t-1) \quad (14)$$

Here, the model parameters $\Lambda, K, \kappa_c, \kappa_{rd}, \upsilon$ are sampled for each patient based on prior distributions from [24]. Additionally, $Rd(t)$ represents the radiation dose applied at time $t$, and $C(t)$ denotes the drug concentration.

We introduce confounding into the assignment of radiotherapy/chemotherapy treatment by making it dependent on past tumor volume evolution. Treatment is simulated using a Bernoulli distribution with probability $\sigma(\pi_t)$, where:

$$\pi_t = \frac{\gamma}{D_{\max}}\left(\bar{D}(t) - \delta\right) \quad (15)$$

In this context: $\bar{D}(t)$ represents the average tumor diameter over the last 15 days, $D_{\max} = 13$ cm is the maximum tumor diameter, $\delta$ is set to $\delta = D_{\max}/2$.

The parameter $\gamma$ controls the level of time-dependent confounding, with a higher $\gamma$ value assigning more weight to the history of tumor diameter in treatment assignment.

## E.2  Additional results

### E.2.1  Comparison to benchmark models

We report in this section detailed counterfactual errors for Causal CPC and baselines over the cancer simulation dataset, which are responsible for Figure 3.

Table 6: Results on the synthetic data set with sequence length 60: mean±standard deviation of NRMSEs. The best value for each metric is given in bold: smaller is better.

| Model | $\tau=1$ | $\tau=2$ | $\tau=3$ | $\tau=4$ | $\tau=5$ | $\tau=6$ | $\tau=7$ | $\tau=8$ | $\tau=9$ | $\tau=10$ |
|---|---|---|---|---|---|---|---|---|---|---|
| | | | | | $\gamma=1$ | | | | | |
| Causal CPC (ours) | **0.83±0.06** | **0.86±0.09** | **0.94±0.09** | **0.97±0.08** | **1.03±0.10** | **1.07±0.10** | **1.12±0.10** | **1.17±0.09** | **1.22±0.08** | **1.26±0.08** |
| CT | 0.99±0.13 | 0.92±0.14 | 0.98±0.14 | 1.05±0.15 | 1.11±0.18 | 1.11±0.11 | 1.21±17 | 1.26±0.16 | 1.31±0.005 | 1.35±0.16 |
| G-Net | 0.91±0.15 | 1.1±0.16 | 1.24±16 | 1.33±0.17 | 1.40±0.18 | 1.47±0.19 | 1.52±0.18 | 1.57±0.22 | 1.63±0.22 | 1.7±0.25 |
| CRN | 0.84±0.10 | 0.83±0.09 | 0.92±0.10 | 1.00±0.11 | 1.09±0.12 | 1.17±0.14 | 1.25±0.16 | 1.32±0.18 | 1.37±0.23 | 1.43±0.26 |
| RMSN | 0.99±0.13 | 0.91±0.04 | 1.30±0.65 | 1.43±0.76 | 1.56±0.83 | 1.73±0.91 | 1.77±0.89 | 1.81±0.88 | 1.84±0.86 | |
| MSM | 1.20±0.10 | 1.83±0.26 | 2.07±0.44 | 2.38±0.44 | 2.54±0.45 | 2.90±0.37 | 3.01±.38 | 3.06±0.36 | 3.08±0.36 | 3.08±0.36 |
| | | | | | $\gamma=2$ | | | | | |
| Causal CPC (ours) | 1.16±0.22 | **0.91±0.10** | **0.95±0.13** | **1.00±0.15** | **1.07±0.19** | **1.17±0.24** | **1.27±0.25** | **1.38±0.28** | **1.49±0.30** | **1.60±0.34** |
| CT | 1.24±0.20 | 1.13±0.15 | 1.27±021 | 1.36±0.28 | 1.44±0.29 | 1.55±0.27 | 1.64±0.28 | 1.69±0.20 | 1.74±0.28 | 1.77±0.29 |
| G-Net | **1.05±0.21** | 1.05±0.08 | 1.26±0.16 | 1.38±0.23 | 1.48±0.27 | 1.57±0.31 | 1.64±0.33 | 1.70±0.36 | 1.75±0.39 | 1.8±0.42 |
| CRN | 1.25±0.25 | 1.08±0.06 | 1.14±0.12 | 1.21±0.17 | 1.30±0.21 | 1.41±0.25 | 1.54±0.32 | 1.67±0.41 | 1.8±0.51 | 1.92±0.63 |
| RMSN | 1.47±0.27 | 1.33±0.25 | 1.30±0.23 | 1.33±0.24 | 1.38±0.26 | 1.45±0.28 | 1.52±0.31 | 1.60±0.25 | 1.67±0.38 | 1.75±0.42 |
| MSM | 1.43±0.27 | 2.22±0.53 | 2.67±0.63 | 2.98±0.70 | 3.19±0.74 | 3.33±0.77 | 3.41±0.79 | 3.44±0.25 | 3.45±0.78 | 3.34±0.77 |
| | | | | | $\gamma=3$ | | | | | |
| Causal CPC (ours) | 1.37±0.31 | **1.16±0.27** | **1.26±0.30** | **1.38±0.35** | **1.53±0.40** | **1.69±0.47** | **1.84±0.52** | **2.00±0.51** | 2.14±0.61 | 2.28±0.66 |
| CT | 1.36±0.32 | 1.42±0.36 | 1.62±0.46 | 1.78±0.53 | 1.89±0.58 | 2.01±0.63 | 2.13±0.66 | 2.22±0.69 | 2.31±0.69 | 2.37±0.73 |
| G-Net | **1.14±0.24** | 1.22±0.15 | 1.54±0.26 | 1.77±0.33 | 1.94±0.36 | 2.09±0.40 | 2.23±0.43 | 2.34±0.47 | 2.44±0.52 | 2.52±0.56 |
| CRN | 1.46±0.29 | 1.54±0.38 | 1.70±0.48 | 1.79±0.53 | 1.86±0.92 | 1.92±0.58 | 1.98±0.59 | 2.04±0.61 | **2.10±0.63** | **2.16±0.64** |
| RMSN | 1.22±0.26 | 1.28±0.29 | 1.43±0.40 | 1.56±0.48 | 1.70±0.53 | 1.83±0.57 | 1.95±0.59 | 2.06±0.61 | 2.14±0.61 | 2.21±0.61 |
| MSM | 1.70±0.35 | 2.73±0.88 | 3.22±1.03 | 3.25±1.12 | 3.71±1.18 | 3.85±1.22 | 3.91±1.23 | 3.95±1.24 | 3.96±1.24 | 3.94±1.23 |

### E.2.2  Ablation study

We detail here the results of the ablation study conducted on the cancer simulation dataset (Table 3). The (full) Causal CPC model, as presented in the core paper, gives, in most cases, better results than any ablation configuration.

Table 7: Results on the synthetic data set with sequence length 40: mean±standard deviation of NRMSEs. The best value for each metric is given in bold: smaller is better.

| Model | $\tau=1$ | $\tau=2$ | $\tau=3$ | $\tau=4$ | $\tau=5$ | $\tau=6$ | $\tau=7$ | $\tau=8$ | $\tau=9$ | $\tau=10$ |
|---|---|---|---|---|---|---|---|---|---|---|
| | | | | | $\gamma=1$ | | | | | |
| Causal CPC (ours) | 1.21±0.07 | **1.13±0.12** | 1.22±0.12 | **1.27±0.17** | **1.35±0.19** | **1.43±0.21** | **1.48±0.22** | **1.54±0.22** | **1.58±0.23** | **1.62±0.23** |
| CT | 1.21±0.09 | 1.34±0.10 | 1.43±0.14 | 1.52±0.19 | 1.59±0.22 | 1.65±0.23 | 1.71±0.23 | 1.76±0.21 | 1.78±0.21 | 1.80±0.18 |
| G-Net | 1.11±0.10 | 1.30±0.17 | 1.52±0.20 | 1.65±0.21 | 1.76±0.24 | 1.86±0.29 | 1.96±0.34 | 2.05±0.40 | 2.13±0.46 | 2.19±0.52 |
| CRN | **1.01±0.12** | **1.11±0.14** | **1.19±0.14** | 1.30±0.13 | 1.41±0.12 | 1.49±0.09 | 1.56±0.07 | 1.62±0.05 | 1.67±0.04 | 1.70±0.04 |
| RMSN | 1.14±0.03 | 1.20±0.08 | 1.24±0.07 | 1.32±0.08 | 1.40±0.08 | 1.49±0.08 | 1.57±0.08 | 1.64±0.06 | 1.71±0.05 | 1.76±0.04 |
| | | | | | $\gamma=2$ | | | | | |
| Causal CPC (ours) | 1.41±0.09 | 1.30±0.17 | 1.33±0.20 | **1.41±0.25** | **1.50±0.28** | **1.56±0.31** | **1.60±0.23** | **1.66±0.27** | **1.70±0.26** | **1.74±0.25** |
| CT | 1.36±0.15 | 1.38±0.19 | 1.40±0.25 | 1.48±0.28 | 1.54±0.30 | 1.62±0.31 | 1.70±0.31 | 1.75±0.31 | 1.82±0.32 | 1.85±0.29 |
| G-Net | 1.26±0.09 | 1.47±0.20 | 1.74±0.28 | 1.90±0.34 | 2.02±0.40 | 2.12±0.44 | 2.22±0.50 | 2.31±0.55 | 2.39±0.61 | 2.47±0.68 |
| CRN | **1.24±0.22** | **1.21±0.18** | **1.31±0.22** | 1.42±0.25 | 1.55±0.29 | 1.67±0.34 | 1.79±0.41 | 1.91±0.50 | 2.03±0.59 | 2.13±0.70 |
| RMSN | 1.28±0.15 | 1.28±0.13 | 1.37±0.12 | 1.49±0.15 | 1.62±0.19 | 1.76±0.25 | 1.89±0.30 | 2.03±0.36 | 2.15±0.41 | 2.26±0.46 |
| | | | | | $\gamma=3$ | | | | | |
| Causal CPC (ours) | 1.52±0.19 | **1.19±0.16** | **1.26±0.23** | **1.38±0.25** | **1.52±0.24** | **1.66±0.25** | **1.79±0.27** | **1.91±0.32** | **2.02±0.38** | **2.11±0.46** |
| CT | 1.56±0.18 | 1.68±0.33 | 1.77±0.54 | 1.95±0.75 | 2.05±0.83 | 2.19±0.92 | 2.29±1.00 | 2.35±1.04 | 2.43±1.09 | 2.43±1.07 |
| G-Net | 1.19±0.02 | 1.38±0.04 | 1.72±0.02 | 1.92±0.08 | 2.11±0.18 | 2.32±0.29 | 2.52±0.41 | 2.72±0.54 | 2.93±0.68 | 3.13±0.83 |
| CRN | 1.50±0.01 | 1.47±0.07 | 1.62±0.24 | 1.76±0.40 | 1.89±0.51 | 1.98±0.57 | 2.05±0.61 | 2.10±0.62 | 2.13±0.61 | 2.15±0.58 |
| RMSN | **1.48±0.19** | 1.40±0.13 | 1.66±0.35 | 1.86±0.52 | 2.01±0.61 | 2.14±0.65 | 2.24±0.67 | 2.32±0.67 | 2.37±0.66 | 2.39±0.65 |

Table 8: Results of the ablation study on the synthetic data set: mean±standard deviation of Normalized Rooted Mean Squared Errors (NRMSEs). The best value for each metric is given in bold: smaller is better.

| Model | $\tau=1$ | $\tau=2$ | $\tau=3$ | $\tau=4$ | $\tau=5$ | $\tau=6$ | $\tau=7$ | $\tau=8$ | $\tau=9$ | $\tau=10$ |
|---|---|---|---|---|---|---|---|---|---|---|
| CAUSAL CPC (FULL) | **0.83 ± 0.06** | **0.86 ± 0.06** | 0.94± 0.09 | **0.97 ± 0.08** | **1.03 ± 0.10** | **1.07±0.10** | **1.12± 0.10** | **1.17± 0.06** | **1.22 ± 0.08** | **1.26± 0.08** |
| Causal CPC (w/o $\mathcal{L}^{(InfoNCE)}$) | 0.84±0.04 | 0.91±0.07 | 0.95± 0.07 | 0.99± 0.09 | 1.03±0.10 | 1.10± 0.07 | 1.15±0.14 | 1.20 ± 0.14 | 1.23±0.14 | 1.28±0.15 |
| Causal CPC (w/o $\mathcal{L}^{(InfoMax)}$) | 0.84±0.04 | 0.86±0.09 | **0.91±0.08** | 0.99±0.10 | 1.07±0.08 | 1.16± 0.08 | 1.24± 0.10 | 1.31 ± 0.12 | 1.38±0.08 | 1.46±0.10 |
| Causal CPC (w CDC loss) | 0.83±0.02 | 0.89±0.07 | 0.96± 0.07 | 1.03± 0.07 | 1.07±0.08 | 1.10± 0.07 | 1.13±0.10 | 1.18 ± 0.09 | 1.24±0.11 | 1.28±0.11 |
| Causal CPC (w Balancing | 0.84±0.04 | 0.88±0.05 | 0.97± 0.05 | 1.04± 0.07 | 1.08±0.10 | 1.13± 0.08 | 1.15±0.14 | 1.20± 0.10 | 1.25±0.08 | 1.29±0.12 |

# F   Experiments on semi-synthetic data: Details

## F.1   Description of the simulation model

In this section, we provide a concise overview of the simulation model built upon the MIMIC III dataset, as introduced by [48]. Initially, a cohort of 1,000 patients is extracted from the MIMIC III data, and the simulation proposed by [48] extends the model of [68].

Let $d_y$ be the dimension of the outcome variable. In the case of multiple outcomes, untreated outcomes, denoted as $\mathbf{Z}_t^{j,(i)}$ for $j = 1, \ldots, d_y$, are generated for each patient $i$ within the cohort. The generation process is defined as follows:

$$\mathbf{Z}_t^{j,(i)} = \underbrace{\alpha_S^j \mathbf{B}\text{-spline}(t) + \alpha_g^j g^{j,(i)}(t)}_{\text{endogenous}} + \underbrace{\alpha_f^j f_Z^j \left( \mathbf{X}_t^{(i)} \right)}_{\text{exogenous}} + \underbrace{\varepsilon_t}_{\text{noise}} \tag{16}$$

where: the B-spline $\mathbf{B}\text{-spline}(t)$ is an endogenous component, $g^{j,(i)}(\cdot)$ is sampled independently for each patient from a Gaussian process with a Matérn kernel and $f_Z^j(\cdot)$ is sampled from a Random Fourier Features (RFF) approximation of a Gaussian process.

To introduce confounding in the assignment mechanism, current time-varying covariates are incorporated via a random function $f_Y^l(\mathbf{X}_t)$ and the average of the subset of the previous $T_l$ treated outcomes, $\bar{A}_{T_l}(\overline{\mathbf{Y}}_{t-1})$. For $d_a$ binary treatments $\mathbf{A}_t^l$, where $l = 1, \ldots, d_a$, the assignment mechanism is modeled as:

$$p_{\mathbf{A}_t^l} = \sigma \left( \gamma_A^l \bar{A}_{T_l} \left( \overline{\mathbf{Y}}_{t-1} \right) + \gamma_X^l f_Y^l \left( \mathbf{X}_t \right) + b_l \right),$$
$$\mathbf{A}_t^l \sim \text{Bernoulli} \left( p_{\mathbf{A}_t^l} \right).$$

Subsequently, treatments are applied to the untreated outcomes using the following expression:

$$E^j(t) = \sum_{i=t-w^l}^t \frac{\min_{l=1,\ldots,d_a} \mathbb{1}_{[\mathbf{A}_i^l=1]} p_{\mathbf{A}_i^l} \beta_{lj}}{\left( w^l - i \right)^2} \tag{17}$$

The final outcome combines the treatment effect and the untreated simulated outcome:

$$Y_t^j = Z_t^j + E^j(t). \tag{18}$$

## F.2 Additional results

### F.2.1 Ablation study

We detail here the results of the ablation study conducted on the MIMIC III semi-synthetic dataset (Table 3). The (full) Causal CPC model, as presented in the core paper, gives consistently better results than any ablation configuration.

Table 9: Results on MIMIC III semi-synthetic data set: mean±standard deviation of Normalized Rooted Mean Squared Errors (NRMSEs). The best value for each metric is given in bold: smaller is better.

| Model | $\tau=1$ | $\tau=2$ | $\tau=3$ | $\tau=4$ | $\tau=5$ | $\tau=6$ | $\tau=7$ | $\tau=8$ | $\tau=9$ | $\tau=10$ |
|---|---|---|---|---|---|---|---|---|---|---|
| **Causal CPC (ful)** | **0.32± 0.04** | **0.45± 0.08** | **0.54±0.06** | **0.61 ± 0.10** | **0.66± 0.10** | **0.69±0.11** | **0.71± 0.11** | **0.73± 0.06** | **0.75 ± 0.05** | **0.77± 0.10** |
| **Causal CPC (w/o $\mathcal{L}^{(InfoNCE)}$)** | 0.35± 0.04 | 0.50± 0.05 | 0.59± 0.06 | 0.66± 0.06 | 0.71± 0.08 | 0.75± 0.06 | 0.77± 0.07 | 0.79± 0.08 | 0.81± 0.07 | 0.83± 0.07 |
| **Causal CPC (w/o $\mathcal{L}^{(InfoMax)}$)** | 0.36± 0.02 | 0.53± 0.03 | 0.64± 0.04 | 0.71± 0.05 | 0.77± 0.05 | 0.77± 0.05 | 0.83± 0.05 | 0.86± 0.05 | 0.88± 0.08 | 0.90± 0.05 |
| **Causal CPC (CDC loss)** | 0.36± 0.02 | 0.54± 0.03 | 0.65± 0.05 | 0.72± 0.05 | 0.77± 0.05 | 0.70± 0.04 | 0.83± 0.04 | 0.85± 0.03 | 0.86± 0.03 | 0.88± 0.08 |
| **Causal CPC (w/o balancing)** | 0.35± 0.03 | 0.50± 0.05 | 0.60± 0.06 | 0.67± 0.06 | 0.72± 0.06 | 0.76± 0.06 | 0.78± 0.06 | 0.80± 0.06 | 0.83± 0.06 | 0.85± 0.06 |

Furthermore, We replace the InfoNCE objective used to compute the CPC term and InfoMax terms with that of NWJ and MINE (Section C.2). We repeat the same MIMIC III experimentation while varying the objective used for CPC and InfoMax. Table 10 shows the counterfactual errors for each configuration compared to the original formulation of Causal CPC. In all cases, The InfoNCE objective performs better with notable error reduction at large horizons.

Table 10: Results of NWJ and MINE MI lower bounds when used for CPC and InfoMax for MIMIC III semi-synthetic data set: mean±standard deviation of Normalized Rooted Mean Squared Errors (NRMSEs). The best value for each metric is given in bold: smaller is better.

| Model | $\tau=1$ | $\tau=2$ | $\tau=3$ | $\tau=4$ | $\tau=5$ | $\tau=6$ | $\tau=7$ | $\tau=8$ | $\tau=9$ | $\tau=10$ |
|---|---|---|---|---|---|---|---|---|---|---|
| **Original Model** | **0.34± 0.04** | **0.45± 0.08** | **0.54±0.06** | **0.61 ± 0.10** | **0.66± 0.10** | **0.69±0.11** | **0.71± 0.11** | **0.73± 0.06** | **0.75± 0.05** | **0.77±0.10** |
| **CPC with NWJ** | 0.34± 0.04 | 0.48± 0.05 | 0.58± 0.06 | 0.66± 0.07 | 0.71± 0.08 | 0.75± 0.07 | 0.78± 0.07 | 0.81± 0.06 | 0.84± 0.06 | 0.87± 0.06 |
| **CPC with MINE** | 0.35± 0.03 | 0.50± 0.05 | 0.61± 0.04 | 0.69± 0.04 | 0.75± 0.04 | 0.79± 0.03 | 0.82± 0.03 | 0.85± 0.02 | 0.88± 0.02 | 0.91± 0.02 |
| **InfoMax with NWJ** | 0.42± 0.08 | 0.56± 0.04 | 0.69± 0.07 | 0.77± 0.08 | 0.83± 0.09 | 0.87± 0.09 | 0.90± 0.09 | 0.92± 0.09 | 0.94± 0.08 | 0.96± 0.08 |
| **InfoMax with MINE** | 0.37± 0.05 | 0.52± 0.03 | 0.65± 0.06 | 0.73± 0.8 | 0.80± 0.10 | 0.84± 0.11 | 0.87± 0.11 | 0.89± 0.10 | 0.91± 0.10 | 0.93± 0.09 |

### F.2.2 Comparison to benchmark models: standard train/test split

As mentioned in Section 6.2, We also tested Causal CPC on MIMIC III semi-synthetic data using the same experimental protocol as [48], namely by using the split of patients into train/validation/test as 800/200/200. As a result, baseline performances in Table 11 are exactly the same as in [48].

Table 11: Results over the MIMIC III semi-synthetic data set (same experimental protocol as in [48]): mean±standard deviation of Rooted Mean Squared Errors (RMSEs). The best value for each metric is given in bold: smaller is better.

| Model | $\tau=1$ | $\tau=2$ | $\tau=3$ | $\tau=4$ | $\tau=5$ | $\tau=6$ | $\tau=7$ | $\tau=8$ | $\tau=9$ | $\tau=10$ |
|---|---|---|---|---|---|---|---|---|---|---|
| **Causal CPC (ours)** | 0.25 ± 0.03 | 0.37 ± 0.02 | **0.40 ± 0.01** | **0.45 ± 0.01** | **0.49 ± 0.02** | **0.52 ± 0.02** | **0.55 ± 0.03** | **0.56 ± 0.03** | **0.58 ± 0.04** | **0.60 ± 0.03** |
| **CT** | **0.20 ± 0.01** | **0.38 ± 0.01** | 0.45 ± 0.01 | 0.49 ± 0.01 | 0.52 ± 0.02 | 0.53 ± 0.02 | **0.55 ± 0.02** | **0.56 ± 0.02** | **0.58 ± 0.02** | **0.59 ± 0.02** |
| **G-Net** | 0.34 ± 0.01 | 0.67 ± 0.03 | 0.83 ± 0.04 | 0.94 ± 0.04 | 1.03 ± 0.05 | 1.10 ± 0.05 | 1.16 ± 0.05 | 1.21 ± 0.06 | 1.25 ± 0.06 | 1.29 ± 0.06 |
| **CRN** | 0.30 ± 0.01 | 0.48 ± 0.02 | 0.59 ± 0.02 | 0.65 ± 0.02 | 0.68 ± 0.02 | 0.71 ± 0.01 | 0.72 ± 0.01 | 0.74 ± 0.01 | 0.76 ± 0.01 | 0.78 ± 0.02 |
| **RMSN** | 0.24 ± 0.01 | 0.47 ± 0.01 | 0.60 ± 0.01 | 0.70 ± 0.02 | 0.78 ± 0.04 | 0.84 ± 0.05 | 0.89 ± 0.06 | 0.94 ± 0.08 | 0.97 ± 0.09 | 1.00 ± 0.11 |
| **MSM** | 0.37 ± 0.01 | 0.57 ± 0.03 | 0.74 ± 0.06 | 0.88 ± 0.03 | 1.14 ± 0.10 | 1.95 ± 1.48 | 3.44 ± 4.57 | > 10.0 | > 10.0 | > 10.0 |

### F.2.3 Running time and model complexity

In this section, we complement the table about complexity and running time given for cancer simulation in the core paper by providing the exact same table but for MIMIC III semi-synthetic data.

## G Proofs of theoretical results

### G.1 Relation between InfoNCE loss and mutual information

**Proposition G.1.**

$$I(\mathbf{U}_{t+j}, \mathbf{C}_t) \geq \log(|\mathcal{B}|) - \mathcal{L}_j^{(InfoNCE)}$$

Table 12: The number of parameters to train for each model after hyper-parameters fine-tuning and the corresponding running time averaged over five seeds. Results are reported for semi-synthetic MIMIC III data; the processing unit is GPU - 1 x NVIDIA Tesla M60 .

| MODEL | TRAINABLE PARAMETERS (K) | TRAINING TIME (MIN) | PREDICTION TIME (MIN) |
|---|---|---|---|
| CAUSAL CPC (OURS) | 9.8 | 12±2 | 4±1 |
| CT | 12 | 14±1 | 38±2 |
| G-NET | 14.7 | 7±1 | 40±3 |
| CRN | 15.1 | 21±2 | 5±1 |
| RMSN | 20 | 48±4 | 5±1 |

*Proof.* In the following, we draw inspiration from the proof of [52]. The InfoNCE loss corresponds to the categorical cross-entropy of classifying the positive sample $\mathbf{U}_{t+j}$ correctly, given the context $\mathbf{C}_t$, with a probability:

$$\frac{\exp(T_j(\mathbf{U}_{t+j}, \mathbf{C}_t))}{\sum_{l=1}^{|\mathcal{B}|} \exp(T_j(\mathbf{U}_{l,t+j}, \mathbf{C}_t))}.$$

The positive sample $\mathbf{U}_{t+j}$ is one element in the batch $\mathcal{B}$, where the remaining elements serve as negative samples. Let $\texttt{pos} \in \{1, \ldots, |\mathcal{B}|\}$ be the indicator of the positive sample $\mathbf{U}_{t+j}$. The optimal probability is given by:

$$p(\text{Index} = \texttt{pos} \mid \mathcal{B}, \mathbf{C}_t) = \frac{p(\mathbf{u}_{\texttt{pos},t+j} \mid \mathbf{C}_t) \prod_{l=1,\ldots,|\mathcal{B}|; l \neq \texttt{pos}} p(\mathbf{u}_{l,t+j})}{\sum_{j=1}^{|\mathcal{B}|} \left[ p(\mathbf{u}_{j,t+j} \mid \mathbf{C}_t) \prod_{l=1,\ldots,|\mathcal{B}|; l \neq j} p(\mathbf{u}_{l,t+j}) \right]} = \frac{\frac{p(\mathbf{u}_{\texttt{pos},t+j}|\mathbf{C}_t)}{p(\mathbf{u}_{\texttt{pos},t+j})}}{\sum_{j=1}^{|\mathcal{B}|} \frac{p(\mathbf{u}_{j,t+j}|\mathbf{C}_t)}{p(\mathbf{u}_{j,t+j})}}.$$

For the score $\exp(T_j(\mathbf{U}_{t+j}, \mathbf{C}_t))$ to be optimal, it should be proportional to $\frac{p(\mathbf{u}_{\texttt{pos},t+j}|\mathbf{C}_t)}{p(\mathbf{u}_{\texttt{pos},t+j})}$. The mutual information (MI) lower bound arises from the fact that $\exp(T_j(\mathbf{U}_{t+j}, \mathbf{C}_t))$ estimates the density ratio $\frac{p(\mathbf{u}_{\texttt{pos},t+j}|\mathbf{C}_t)}{p(\mathbf{u}_{\texttt{pos},t+j})}$.

$$\begin{aligned}
\mathcal{L}_j^{(InfoNCE)} &= -\mathbb{E}_{\mathcal{B}} \log \left[ \frac{\frac{p(\mathbf{u}_{t+j}|\mathbf{c}_t)}{p(\mathbf{u}_{t+j})}}{\frac{p(\mathbf{u}_{t+j}|\mathbf{c}_t)}{p(\mathbf{u}_{t+j})} + \sum_{\mathbf{u}_{l,t+j} \in \mathcal{B}_{\text{neg}}} \frac{p(\mathbf{u}_{l,t+j}|\mathbf{c}_t)}{p(\mathbf{u}_{l,t+j})}} \right] \\
&= \mathbb{E}_{\mathcal{B}} \log \left[ 1 + \frac{p(\mathbf{u}_{t+j})}{p(\mathbf{u}_{t+j} \mid \mathbf{c}_t)} \sum_{\mathbf{u}_{l,t+j} \in \mathcal{B}_{\text{neg}}} \frac{p(\mathbf{u}_{l,t+j} \mid \mathbf{c}_t)}{p(\mathbf{u}_{l,t+j})} \right] \\
&\approx \mathbb{E}_{\mathcal{B}} \log \left[ 1 + \frac{p(\mathbf{u}_{t+j})}{p(\mathbf{u}_{l,t+j} \mid \mathbf{c}_t)} (|\mathcal{B}| - 1) \mathbb{E}_{\mathbf{U}_{t+j}} \frac{p(\mathbf{u}_{l,t+j} \mid \mathbf{c}_t)}{p(\mathbf{u}_{l,t+j})} \right] \qquad (19) \\
&= \mathbb{E}_{\mathcal{B}} \log \left[ 1 + \frac{p(\mathbf{u}_{t+j})}{p(\mathbf{u}_{t+j} \mid \mathbf{c}_t)} (|\mathcal{B}| - 1) \right] \\
&\geq \mathbb{E}_{\mathcal{B}} \log \left[ \frac{p(\mathbf{u}_{t+j})}{p(\mathbf{u}_{t+j} \mid \mathbf{c}_t)} |\mathcal{B}| \right] \\
&= -I(\mathbf{u}_{t+j}, \mathbf{c}_t) + \log(|\mathcal{B}|),
\end{aligned}$$

The approximation in the third equation, Eq. (19), becomes more precise as the batch size increases. □

## G.2 Relation between InfoMax and Input Reconstruction

We now prove Proposition 5.1, which states that:

$$I(\mathbf{C}_t^h, \mathbf{C}_t^f) \leq I(\mathbf{H}_t, (\mathbf{C}_t^h, \mathbf{C}_t^f)).$$

*Proof.* This follows from two applications of the data processing inequality [17], which states that for random variables $A$, $B$, and $C$ satisfying the Markov relation $A \to B \to C$, the inequality $I(A; C) \leq I(A; B)$ holds.

First, since $\mathbf{C}_t^h = \Phi_{\theta_1, \theta_2}(\mathbf{H}_t^h)$ and $\mathbf{C}_t^f = \Phi_{\theta_1, \theta_2}(\mathbf{H}_t^f)$, we can write $\mathbf{H}_t^h = \text{trunc}_f(\mathbf{H}_t)$ and $\mathbf{H}_t^f = \text{trunc}_h(\mathbf{H}_t)$, where $\text{trunc}_f$ and $\text{trunc}_h$ truncate the future and history processes, respectively, given a splitting time $t_0$.

Thus, we have the Markov relation:

$$\mathbf{C}_t^h \xleftarrow{\Phi_{\theta_1,\theta_2} \circ \mathrm{trunc}_f} \mathbf{H}_t \xrightarrow{\Phi_{\theta_1,\theta_2} \circ \mathrm{trunc}_h} \mathbf{C}_t^f,$$

which is Markov equivalent to:

$$\mathbf{C}_t^h \xrightarrow{\Phi_{\theta_1,\theta_2} \circ \mathrm{trunc}_f} \mathbf{H}_t \xrightarrow{\Phi_{\theta_1,\theta_2} \circ \mathrm{trunc}_h} \mathbf{C}_t^f.$$

By the data processing inequality, this results in $I(\mathbf{C}_t^h, \mathbf{C}_t^f) \leq I(\mathbf{H}_t, \mathbf{C}_t^h)$. On the other hand, we have the trivial Markov relation $\mathbf{H}_t \to (\mathbf{C}_t^h, \mathbf{C}_t^f) \to \mathbf{C}_t^h$, which implies $I(\mathbf{H}_t, \mathbf{C}_t^h) \leq I(\mathbf{H}_t, (\mathbf{C}_t^h, \mathbf{C}_t^f))$. Combining these two inequalities proves the proposition. $\qquad\square$

### G.3 Proof of Theorem 5.2

To begin, we split the process history into two non-overlapping views (Figure 2): $\mathbf{H}_t^h := \mathbf{U}_{1:t_0}$ and $\mathbf{H}_t^f := \mathbf{U}_{t_0+1:t}$, representing a historical subsequence and a future subsequence within the process history $\mathbf{H}_t$, respectively. We then computed representations of these two views denoted $\mathbf{C}_t^h$ and $\mathbf{C}_t^f$, respectively. This naturally gives rise to the Markov chain, as in showed in the proof of proposition 5.1:

$$\mathbf{C}_t^h \leftarrow \mathbf{H}_t \to \mathbf{C}_t^f$$

which is Markov equivalent to:

$$\mathbf{C}_t^h \to \mathbf{H}_t \to \mathbf{C}_t^f$$

Following this Markov chain, we can show that [71]:

$$I(\mathbf{C}_t^f, \mathbf{C}_t^h) = I(\mathbf{H}_t, \mathbf{C}_t^h) - \mathbb{E}_{\mathbf{h}_t \sim \mathbb{P}_{\mathbf{H}_t}} \mathbb{E}_{\mathbf{c}_t^f \sim \mathbb{P}_{\mathbf{C}_t^f | \mathbf{h}_t}} \left[ D_{KL}[\mathbb{P}_{\mathbf{C}_t^h | \mathbf{h}_t} || \mathbb{P}_{\mathbf{C}_t^h | \mathbf{c}_t^f}] \right]$$

On the other hand, by applying the chain rule of the mutual information to $I(\mathbf{H}_t; (\mathbf{C}_t^h, \mathbf{C}_t^f))$ we get:

$$I(\mathbf{H}_t; (\mathbf{C}_t^f, \mathbf{C}_t^h)) = I(\mathbf{H}_t, \mathbf{C}_t^h) + I(\mathbf{H}_t; \mathbf{C}_t^f \mid \mathbf{C}_t^h)$$

Combining these equations, the tightness of our bounds can be written as:

$$I(\mathbf{H}_t; (\mathbf{C}_t^f, \mathbf{C}_t^h)) - I(\mathbf{C}_t^f, \mathbf{C}_t^h) = I(\mathbf{H}_t; \mathbf{C}_t^f \mid \mathbf{C}_t^h) + \mathbb{E}_{\mathbf{h}_t \sim \mathbb{P}_{\mathbf{H}_t}} \mathbb{E}_{\mathbf{c}_t^f \sim \mathbb{P}_{\mathbf{C}_t^f | \mathbf{h}_t}} \left[ D_{KL}[\mathbb{P}_{\mathbf{C}_t^h | \mathbf{h}_t} || \mathbb{P}_{\mathbf{C}_t^h | \mathbf{c}_t^f}] \right]$$

### G.4 On the Relation Between Conditional Entropy and Reconstruction

We now prove the statement in the core paper, which asserts that the conditional entropy $H(\mathbf{H}_t \mid (\mathbf{C}_t^h, \mathbf{C}_t^f)) \geq 0$ is minimized if $\mathbf{H}_t$ is almost surely a function of $(\mathbf{C}_t^h, \mathbf{C}_t^f)$. The proof is adapted from [17].

**Proposition G.2.** *If $H(\mathbf{A} \mid \mathbf{B}) = 0$, then $\mathbf{A} = f(\mathbf{B})$ almost surely.*

*Proof.* For simplicity, suppose $\mathbf{A}$ and $\mathbf{B}$ are discrete random variables. Assume, by contradiction, that there exists $\mathbf{b}_0$ and two distinct values $\mathbf{a}_1$ and $\mathbf{a}_2$ such that $p(\mathbf{a}_1 \mid \mathbf{b}_0) > 0$ and $p(\mathbf{a}_2 \mid \mathbf{b}_0) > 0$. Then, the conditional entropy is given by:

$$H(\mathbf{A} \mid \mathbf{B}) = -\sum_{\mathbf{b}} p(\mathbf{b}) \sum_{\mathbf{a}} p(\mathbf{a} \mid \mathbf{b}) \log p(\mathbf{a} \mid \mathbf{b}).$$

In particular, we have:

$$H(\mathbf{A} \mid \mathbf{B}) \geq p(\mathbf{b}_0) \left( -p(\mathbf{a}_1 \mid \mathbf{b}_0) \log p(\mathbf{a}_1 \mid \mathbf{b}_0) - p(\mathbf{a}_2 \mid \mathbf{b}_0) \log p(\mathbf{a}_2 \mid \mathbf{b}_0) \right) > 0.$$

Since $-t \log t \geq 0$ for $0 \leq t \leq 1$ and is strictly positive for $t$ not equal to 0 or 1, the conditional entropy $H(\mathbf{A} \mid \mathbf{B}) = 0$ if and only if $\mathbf{A}$ is almost surely a function of $\mathbf{B}$. $\qquad\square$

## G.5 On the benefit of the InfoMax loss on inverting the data generation process

To ensure identifiability in the latent space, we leverage recent advances in causal and disentangled representation learning. Suppose the true data-generating process is given by $\mathbf{H}_t = g(\mathbf{z}_t)$, where $\mathbf{z}_t$ represents the true latent factors. In the sequential context, we assume that the same function $g$ generates two historical subsequences:

$$\mathbf{H}_t^f = g(\mathbf{z}_t^f), \quad \mathbf{H}_t^h = g(\mathbf{z}_t^h).$$

We assume a general dependency of the form:

$$p(\mathbf{z}_t^f \mid \mathbf{z}_t^h) = \frac{Q(\mathbf{z}_t^f)}{Z(\mathbf{z}_t^h)} \exp(-d(\mathbf{z}_t^f, \mathbf{z}_t^h)).$$

Here, $\mathbf{\Phi}$ is an encoder, and we use the InfoMax regularization term as follows:

$$\mathcal{L}^{(InfoMax)}(\mathbf{\Phi}, d, \mathcal{B}) := -\mathbb{E}_{\mathcal{B}} \left[ \log \frac{\exp(-d(\mathbf{\Phi}(\mathbf{H}_t^f), \mathbf{\Phi}(\mathbf{H}_t^h)))}{\sum_{l=1}^{|\mathcal{B}|} \exp(-d(\mathbf{\Phi}(\mathbf{H}_{l,t}^f), \mathbf{\Phi}(\mathbf{H}_t^h)))} \cdot \right]$$

According to [47], under certain conditions, if the encoder $f$ minimizes $\mathcal{L}^{(InfoMax)}$, then $h = g \circ f$ is a scaled permutation matrix. This result suggests that when the encoder achieves a minimizer for $\mathcal{L}^{(InfoMax)}$, the encoder function $f$ closely approximates an invertible transformation of $g$.

From a causal inference perspective, if $Y_{it}(\omega_{it}) \perp\!\!\!\perp W_{it} \mid \mathbf{H}_{it}$ and $\mathbf{H}_{it} = g(\mathbf{Z}_{it})$, then an invertible function $g \circ f$ ensures that:

$$Y_{it}(\omega_{it}) \perp\!\!\!\perp W_{it} \mid g \circ f(\mathbf{H}_{it}).$$

Thus, $Y_{it}(\omega_{it}) \perp\!\!\!\perp W_{it} \mid g(\mathbf{C}_{it})$ and since $g$ is invertible, we have:

$$Y_{it}(\omega_{it}) \perp\!\!\!\perp W_{it} \mid \mathbf{C}_{it}.$$

This demonstrates that the representation $\mathbf{C}_{it}$ retains the essential independence structure, facilitating accurate counterfactual inference.

## G.6 Proof of theorem 5.4

To prove the Theorem 5.4, we first prove the following lemma and proposition.

**Lemma G.3.** *Let $\mathbf{\Phi}$ be a fixed representation function. Given that $q(W_{t+1} \mid \mathbf{\Phi}(\mathbf{H}_t))$ is the conditional likelihood of observing the treatment $W_{t+1}$, denote the probability of observing each treatment value as $q^j = q(\mathbf{\Phi}(\mathbf{H}_t)) := q(W_{t+1} = j \mid \mathbf{\Phi}(\mathbf{H}_t))$ for $j \in \{0, 1, \ldots, K-1\}$. Then, the optimal treatment prediction function is such that*

$$q^{j,*}(\mathbf{\Phi}(\mathbf{H}_t)) = \frac{p(\mathbf{\Phi}(\mathbf{H}_t) \mid W_{t+1} = j)}{\sum_{l=0}^{K-1} p(\mathbf{\Phi}(\mathbf{H}_t) \mid W_{t+1} = l) p(W_{t+1} = l)} \tag{20}$$

*Proof.* For a fixed representation $\mathbf{\Phi}$, finding the optimal treatment probabilities amounts to solving the following optimization problem:

$$\max_q \mathbb{E}_{\mathbb{P}(\mathbf{\Phi}(\mathbf{H}_t), W_{t+1})} \left[ \log q(W_{t+1} \mid \mathbf{\Phi}(\mathbf{H}_t)) \right] \quad \text{subject to} \quad \sum_{l=0}^{K-1} q^l(\mathbf{\Phi}(\mathbf{H}_t)) = 1 \tag{21}$$

First, we write the likelihood $q(W_{t+1} \mid \mathbf{\Phi}(\mathbf{H}_t))$ using the conditional probabilities $q^j(\mathbf{\Phi}(\mathbf{H}_t))$.

$$q(W_{t+1} \mid \mathbf{\Phi}(\mathbf{H}_t)) = \prod_{j=0}^{K-1} q^j(\mathbf{\Phi}(\mathbf{H}_t))^{\mathbb{1}\{W_{t+1}=j\}}$$

Then, the treatment likelihood can be written as

$$\mathbb{E}_{\mathbb{P}(\Phi(\mathbf{H}_t),W_{t+1})}\left[\log q(W_{t+1}\mid\Phi(\mathbf{H}_t))\right] = \mathbb{E}_{\mathbb{P}(\Phi(\mathbf{H}_t),W_{t+1})}\left[\sum_{l=0}^{K-1}\log(q^l(\Phi(\mathbf{H}_t)))\mathbb{1}_{\{W_{t+1}=j\}}\right]$$

$$= \sum_{l=0}^{K-1}\int \log(q^l(\Phi(\mathbf{H}_t))\mathbb{1}_{\{W_{t+1}=j\}}p(W_{t+1}\mid\Phi(\mathbf{H}_t))p(\Phi(\mathbf{H}_t))dW_{t+1}d\Phi(\mathbf{H}_t)$$

$$= \sum_{l=0}^{K-1}\int \log(q^l(\Phi(\mathbf{H}_t))p(W_{t+1}=l\mid\Phi(\mathbf{H}_t))p(\Phi(\mathbf{H}_t))d\Phi(\mathbf{H}_t)$$

$$= \sum_{l=0}^{K-1}\int \log(q^l(\Phi(\mathbf{H}_t))p(\Phi(\mathbf{H}_t)\mid W_{t+1}=l)p(W_{t+1}=l)d\Phi(\mathbf{H}_t)$$

Let's denote $\alpha_l = p(W_{t+1}=l)$, the marginal probability of observing the $l$-th treatment regime, and $p_l^\Phi(\mathbf{H}_t) = p(\Phi(\mathbf{H}_t)\mid W_{t+1}=l)$ with a corresponding probability distribution $\mathbb{P}_l^\Phi$. We intend to maximize point-wise the objective in Eq. (21). Plugging the latter formulation of the conditional likelihood in Eq. (21) and writing the Lagrangian function, we get

$$\max_q \sum_{l=0}^{K-1}\log(q^l(\Phi(\mathbf{H}_t))p_j^\Phi(\mathbf{H}_t)\alpha_l + \lambda(\sum_{l=0}^{K-1}q^l(\Phi(\mathbf{H}_t))-1) \tag{22}$$

Computing the gradient w.r.t $q^l(\Phi(\mathbf{H}_t))$ for $l\in\{0,1,\ldots,K-1\}$ and setting to zero, we have

$$q^{l,*}(\Phi(\mathbf{H}_t)) = -\frac{\alpha_l p_j^\Phi(\mathbf{H}_t)}{\lambda} \tag{23}$$

Then, by the equality constraint, we find that $\lambda = -\sum_{l=0}^{K-1}\alpha_l p_j^\Phi(\mathbf{H}_t)$. $\square$

**Proposition G.4.** *Let $\Phi$ be a fixed representation function. The $I_{CLUB}$ objective when the treatment prediction function is optimal (i.e. $q = q^*$) has the following form:*

$$I_{CLUB} = \sum_{j=0}^{K-1}\alpha_l D_{KL}(\mathbb{P}_j^\Phi\|\sum_{l=0}^{K-1}\alpha_l\mathbb{P}_l^\Phi) + \mathbb{E}_{\mathbb{P}_{\Phi(\mathbf{H}_t)}}\left[D_{KL}(\mathbb{P}_{W_{t+1}}\|\mathbb{P}_{W_{t+1}\mid\Phi(\mathbf{H}_t)})\right]. \tag{24}$$

*Proof.* First, recall that

$$I_{\text{CLUB}}(\Phi(\mathbf{H}_t),W_{t+1};q^*) = \mathbb{E}_{\mathbb{P}_{(\Phi(\mathbf{H}_{t+1}),W_{t+1})}}\left[\log q^*(W_{t+1}\mid\Phi(\mathbf{H}_{t+1}))\right] - \mathbb{E}_{\mathbb{P}_{\Phi(\mathbf{H}_{t+1})}}\mathbb{E}_{\mathbb{P}_{W_{t+1}}}\left(\log q^*(W_{t+1}\mid\Phi(\mathbf{H}_{t+1})))\right]$$

$$I_{\text{CLUB}}(\Phi(\mathbf{H}_t),W_{t+1};q^*) = A - B$$

Let's detail $A$ and $B$ separately,

$$A = \sum_{j=0}^{K-1}\int \alpha_j\log(q^{l,*}(\Phi(\mathbf{H}_t))p_j^\Phi(\mathbf{H}_t)d\Phi(\mathbf{H}_t)$$

$$= \sum_{j=0}^{K-1}\int \alpha_j\log(\frac{\alpha_j p_j^\Phi(\mathbf{H}_t)}{\sum_{l=0}^{K-1}p_l^\Phi(\mathbf{H}_t)\alpha_l})p_j^\Phi(\mathbf{H}_t)d\Phi(\mathbf{H}_t)$$

$$= \sum_{j=0}^{K-1}\int \alpha_j\log(\frac{p_j^\Phi(\mathbf{H}_t)}{\sum_{l=0}^{K-1}p_l^\Phi(\mathbf{H}_t)\alpha_l})p_j^\Phi(\mathbf{H}_t)d\Phi(\mathbf{H}_t) + \log(\alpha_j)\alpha_j$$

$$= \sum_{j=0}^{K-1}\alpha_j D_{KL}(\mathbb{P}_j^\Phi\|\sum_{l=0}^{K-1}\alpha_l\mathbb{P}_l^\Phi) + \sum_{j=0}^{K-1}\log(\alpha_j)\alpha_j$$

Finally, we can write

$$A = \sum_{j=0}^{K-1}\alpha_j D_{KL}(\mathbb{P}_j^\Phi\|\sum_{l=0}^{K-1}\alpha_l\mathbb{P}_l^\Phi) - H(W_{t+1}) \tag{25}$$

For the remaining term $B$, we have

$$
\begin{aligned}
B &= \mathbb{E}_{\mathbb{P}_{\Phi(\mathbf{H}_t)}} \mathbb{E}_{\mathbb{P}_{W_{t+1}}} \left( \log q^*(W_{t+1} \mid \Phi(\mathbf{H}_{t+1})) \right) \\
&= \sum_{j=0}^{K-1} \mathbb{E}_{\mathbb{P}_{\Phi(\mathbf{H}_t)}} \mathbb{E}_{\mathbb{P}_{W_{t+1}}} \left[ \log(q^j(\Phi(\mathbf{H}_t))) \mathbb{1}_{\{W_{t+1}=j\}} \right] \\
&= \sum_{j=0}^{K-1} \mathbb{E}_{\mathbb{P}_{\Phi(\mathbf{H}_t)}} \left[ \alpha_j \log(q^j(\Phi(\mathbf{H}_t))) \right] \\
&= \sum_{j=0}^{K-1} \alpha_j \int \log \left[ \frac{\alpha_j p_j^{\Phi}(\mathbf{H}_t)}{\sum_{l=0}^{K-1} p_l^{\Phi}(\mathbf{H}_t)\alpha_l} \right] p(\Phi(\mathbf{H}_t)) d\Phi(\mathbf{H}_t) \\
&= \sum_{j=0}^{K-1} \alpha_j \int \log \left[ \frac{p(\Phi(\mathbf{H}_t))}{\sum_{l=0}^{K-1} p_l^{\Phi}(\mathbf{H}_t)\alpha_l} \frac{p(W_{t+1} = j \mid \Phi(\mathbf{H}_t))}{p(W_{t+1} = j)} \right] p(\Phi(\mathbf{H}_t)) d\Phi(\mathbf{H}_t) \\
&\quad - H(W_{t+1}) \\
&= \sum_{j=0}^{K-1} \alpha_j \int \underbrace{\log \left[ \frac{p(\Phi(\mathbf{H}_t))}{\sum_{l=0}^{K-1} p_l^{\Phi}(\mathbf{H}_t)\alpha_l} \right] p(\Phi(\mathbf{H}_t)) d\Phi(\mathbf{H}_t)}_{=0} \\
&\quad + \sum_{j=0}^{K-1} \alpha_j \int \log \left[ \frac{p(W_{t+1} = j \mid \Phi(\mathbf{H}_t))}{p(W_{t+1} = j)} \right] p(\Phi(\mathbf{H}_t)) d\Phi(\mathbf{H}_t) - H(W_{t+1})
\end{aligned}
$$

The final form of $B$ is therefore

$$
B = -\int D_{KL}(\mathbb{P}_{W_{t+1}} | \mathbb{P}_{W_{t+1}|\Phi(\mathbf{H}_t)}) p(\Phi(\mathbf{H}_t)) d\Phi(\mathbf{H}_t) - H(W_{t+1}) \tag{26}
$$

The proposition follows immediately from Equations (25) and (26). $\qquad\square$

*Proof.* (Theorem 5.4) Since by lemma G.3, the $I_{CLUB}$ formulation in proposition G.4 holds, then to prove that the representation is balanced, it is enough to see that by the positivity of $D_{KL}$

$$
I_{CLUB} \geq \mathbb{E}_{\mathbb{P}_{\Phi(\mathbf{H}_t)}} \left[ D_{KL}(\mathbb{P}_{W_{t+1}} || \mathbb{P}_{W_{t+1}|\Phi(\mathbf{H}_t)}) \right] \geq 0 \tag{27}
$$

$I_{CLUB}$ is minimal when $I_{CLUB} = 0$, which happens if and only if for $j \in \{0, 1, \ldots, K-1\}$ $p(W_{t+1} = j) = p(W_{t+1} = j \mid \Phi(\mathbf{H}_t))$ almost surely which, by Bayes rule is equivalent to say $p(\Phi(\mathbf{H}_t)) = p(\Phi(\mathbf{H}_t) \mid W_{t+1} = j)$.

$\qquad\square$

## H  Causal CPC Pseudo algorithm

In this section, we present a detailed overview of the training procedure for Causal CPC. Initially, we train the Encoder using only the contrastive terms, as outlined in Algorithm 1. Our primary objective is to ensure that, for each time step $t$, the process history $\mathbf{H}_t$ is predictive of future local features $\mathbf{Z}_t$. However, calculating the InfoNCE loss for a batch across all possible time steps $t = 0, \ldots, t_{\max}$ can be computationally demanding.

To address this, we adopt a more efficient approach by uniformly sampling a single time step $t$ per batch. Subsequently, the corresponding process history $\mathbf{H}_t$ is contrasted. The sampled $\mathbf{H}_t$ is then employed as input for the InfoMax objective and randomly partitioned into future $\mathbf{H}_t^f$ and past $\mathbf{H}_t^h$ sub-processes.

The decoder is trained while taking the encoder as input (Algorithm 2), utilizing a lower learning rate compared to the untrained part of the decoder. It is trained autoregressively and without teacher forcing. This implies that for each time step $t$, our GRU-based decoder should predict the future sequence of treatments $\hat{Y}_{t+1:t+\tau}$ with its hidden state initialized to the representation $\Phi_t$ of the historical process up to time $t$.

---
**Algorithm 1** Pretraining of the encoder
___

**Require:** Encoder parameters $\theta_{1,2,3}$, learning rate $\mu$
  **Input:** data $\{\mathbf{H}_{i,t_{max}}, i = 1, \ldots, N\}$
  **for** $p \in \{1, \ldots, \text{epoch}_{max}\}$ **do**
    **for** $\mathcal{B} = \{\mathbf{H}_{i,t_{max}}, i = 1, \ldots, |\mathcal{B}|\}$ **do**
      $\mathbf{Z}_t = \Phi_{\theta_1}([\mathbf{X}_t, W_{t-1}, Y_{t-1}])$ for $t = 0, \ldots, t_{max}$.
      Choose $t \sim \mathcal{U}([1, t_{max} - 1])$.
      Compute $\mathbf{C}_t = \Phi_{\theta_1, \theta_2}(\mathbf{H}_t)$.
      Compute $\mathcal{L}^{CPC}(\theta_1, \theta_2, \{\Gamma_j\}_{j=1}^{\tau})$.
      Choose $t_0 \sim \mathcal{U}([1, t])$.
      Compute $\mathbf{C}_t^h = \Phi_{\theta_1, \theta_2}(\mathbf{H}_t^h)$, $\mathbf{C}_t^f = \Phi_{\theta_1, \theta_2}(\mathbf{H}_t^f)$,
      Compute $\mathcal{L}^{(InfoMax)}(\theta_1, \theta_2, \gamma)$.
      Update parameters

$$\theta_{1,2,3} \leftarrow \theta_{1,2,3} - \mu \left( \frac{\partial \mathcal{L}^{CPC}(\theta_1, \theta_2, \{\Gamma_j\}_{j=1}^{\tau})}{\partial \theta_{1,2,3}} + \frac{\partial \mathcal{L}^{(InfoMax)}(\theta_1, \theta_2, \gamma)}{\partial \theta_{1,2,3}} \right)$$

    **end for**
  **end for**
  **Return**: Trained encoder.
___

To enhance training efficiency, instead of predicting $\hat{Y}_{i,t+1:t+\tau}$ for all individuals $i$ in a batch and for all possible time steps $t$, we randomly select $m$ time indices $t_{i,1}, \ldots, t_{i,m}$ for each individual $i$. From these indices, we compute future treatment response sequences $\hat{Y}_{i,t_{i,1}+1:t_{i,1}+\tau}, \ldots, \hat{Y}_{i,t_{i,m}+1:t_{i,m}+\tau}$. We found that is enough to train while selecting randomly 10% of the time steps.

---
**Algorithm 2** Training of the decoder
___

**Require:** Encoder parameters $\theta_{1,2,3}$, Decoder parameters $\theta_4, \theta_Y, \theta_W$.
**Require:** Encoder learning rate $\mu_{enc}$, Treatment learning rate $\mu_W$, Outcome learning rate $\mu_Y$.
**Require:** Number of random time indices $m$.
  **Input:** data $\{\mathbf{H}_{i,t_{max}}, i = 1, \ldots, N\}$
  **for** $p \in \{1, \ldots, \text{epoch}_{max}\}$ **do**
    **for** $\mathcal{B} = \{\mathbf{H}_{i,t_{max}}, i = 1, \ldots, |\mathcal{B}|\}$ **do**
      Compute $\mathbf{C}_{i,t} = \text{encoder}(\mathbf{H}_{i,t})$ for $t = 0, \ldots, t_{max}, i = 1, \ldots, |\mathcal{B}|$.
      Compute $\mathbf{\Phi}_t = \mathbf{\Phi}_{\theta_R}(\mathbf{H}_t)$.
      **for** $i = 1, \ldots, |\mathcal{B}|$ **do**
        Choose $t_{i,1}, \ldots, t_{i,m} \sim \mathcal{U}([1, t_{max} - \tau])$.
        **for** $t \in \{t_{i,1}, \ldots, t_{i,m}\}$ **do**
          Compute $\hat{Y}_{i,t+1:t+\tau}, \hat{W}_{i,t+1:t+\tau}, \mathbf{\Phi}_{i,t+1:t+\tau-1} = \text{decoder}(\mathbf{\Phi}_t, \mathbf{V}_i, W_{i,t}, Y_{i,t}, W_{i,t+1:t+\tau})$
        **end for**
      **end for**
      Compute $\mathcal{L}_{dec}(\theta_R, \theta_Y, \theta_W)$ and $\mathcal{L}_W(\theta_W, \theta_R)$.
      Update parameters in the order.

$$\theta_{1,2,3} \leftarrow \theta_{1,2,3} - \mu_{enc} \left( \frac{\partial \mathcal{L}_{dec}(\theta_R, \theta_Y, \theta_W)}{\partial \theta_{1,2,3}} \right)$$

$$\theta_{4,Y} \leftarrow \theta_{4,Y} - \mu_Y \left( \frac{\partial \mathcal{L}_{dec}(\theta_R, \theta_Y, \theta_W)}{\partial \theta_{4,Y}} \right)$$

$$\theta_W \leftarrow \theta_W - \mu_W \left( \frac{\partial \mathcal{L}_W(\theta_W, \theta_R)}{\partial \theta_W} \right)$$

    **end for**
  **end for**
  **Return**: Trained decoder.
___

# I  Causal CPC: Architecture details

| |
|---|
| **Inputs**: $[\mathbf{X}_t, W_{t-1}, Y_{t-1}]$ |
| Linear Layer |
| WeightNorm |
| SELU |
| Linear Layer |
| WeightNorm |
| **Outputs**: $\mathbf{Z}_t = \Phi_{\theta_1}([\mathbf{X}_t, W_{t-1}, Y_{t-1}])$ |

Table 13: Architecture for learning local features $\mathbf{Z}_t$

| |
|---|
| **Inputs**: $\mathbf{Z}_{\leq t}$ |
| GRU (1 layer) |
| **Outputs**: Hidden state $\mathbf{C}_t = \Phi_{\theta_2}^{ar}(\mathbf{Z}_{\leq t})$ |

Table 14: Architecture for learning context representation $\mathbf{C}_t$

| |
|---|
| **Inputs**: $[\mathbf{\Phi}_t, W_t]$ |
| Linear Layer |
| WeightNorm |
| SELU |
| Linear Layer |
| WeightNorm |
| **Outputs**: $\hat{Y}_t$ |

Table 15: Architecture for outcome prediction

| |
|---|
| **Inputs**: $\mathbf{\Phi}_t$ |
| Linear Layer |
| SpectralNorm |
| SELU |
| Linear Layer |
| SpectralNorm |
| **Outputs**: $\hat{W}_t$ |

Table 16: Architecture for treatment prediction

# J  Models hyperparameters

In this section, we report the range of all hyperparameters to be fine-tuned, as well as fixed hyperparameters for all models and across the different datasets used in experiments. Best hyperparameter values are reported in the configuration files in the code repository.

Table 17: Hyper-parameters search range for RMSN

| Model | Sub-model | Hyperparameter | Cancer simulation | MIMIC III (SS) |
|---|---|---|---|---|
| RMSNs | Propensity Treatment Network | LSTM layers | 1 | 1 |
| | | Learning rate | $0.01, 0.005, 0.001, 0.0001$ | $0.01, 0.005, 0.001, 0.0001$ |
| | | Batch size | $32, 64, 128$ | $32, 64, 128$ |
| | | LSTM hidden units | $4, 6, \ldots, 12$ | $4, 6, \ldots, 30$ |
| | | LSTM dropout rate | - | - |
| | | Max gradient norm | $0.5, 1, 2$ | $0.5, 1, 2$ |
| | | Early Stopping (min delta) | 0.0001 | 0.0001 |
| | | Early Stopping (patience) | 30 | 30 |
| Propensity History Network | | LSTM layers | 1 | 1 |
| | | Learning rate | $0.01, 0.005, 0.001, 0.0001$ | $0.01, 0.005, 0.001, 0.0001$ |
| | | Batch size | $32, 64, 128$ | $64, 128, 256$ |
| | | LSTM hidden units | $4, 6, \ldots, 20$ | $4, 6, \ldots, 30$ |
| | | LSTM dropout rate | - | - |
| | | Early Stopping (min delta) | 0.0001 | 0.0001 |
| | | Early Stopping (patience) | 30 | 30 |
| Encoder | | LSTM layers | 1 | 1 |
| | | Learning rate | $0.01, 0.005, 0.001, 0.0001$ | $0.01, 0.005, 0.001, 0.0001$ |
| | | Batch size | $32, 64, 128$ | $32, 64, 128$ |
| | | LSTM hidden units | $4, 6, \ldots, 20$ | $4, 6, \ldots, 30$ |
| | | LSTM dropout rate | - | - |
| | | Early Stopping (min delta) | 0.0001 | 0.0001 |
| | | Early Stopping (patience) | 30 | 30 |
| Decoder | | LSTM layers | 1 | 1 |
| | | Learning rate | $0.01, 0.005, 0.001, 0.0001$ | $0.01, 0.005, 0.001, 0.0001$ |
| | | Batch size | $32, 64, 128$ | $128, 512, 1024$ |
| | | LSTM hidden units | $4, 6, \ldots, 20$ | $4, 6, \ldots, 30$ |
| | | LSTM dropout rate | - | - |
| | | Max gradient norm | $0.5, 1, 2$ | $0.5, 1, 2$ |
| | | Early Stopping (min delta) | 0.0001 | 0.0001 |
| | | Early Stopping (patience) | 30 | 30 |

Table 18: Hyper-parameters search range for CRN

| Model | Sub-model | Hyperparameter | Cancer simulation | MIMIC III (SS) |
|---|---|---|---|---|
| CRN | Encoder | LSTM layers | 1 | 1 |
| | | Learning rate | $0.01, 0.005, 0.001, 0.0001$ | $0.01, 0.005, 0.001, 0.0001$ |
| | | Batch size | $32, 64, 128$ | $32, 64, 128$ |
| | | LSTM hidden units | $4, 6, \ldots, 30$ | $4, 6, \ldots, 30$ |
| | | LSTM dropout rate | - | - |
| | | BR size | $4, 6, \ldots, 20$ | $4, 6, \ldots, 30$ |
| | | Early Stopping (min delta) | 0.0001 | 0.0001 |
| | | Early Stopping (patience) | 30 | 30 |
| | Decoder | LSTM layers | 1 | 1 |
| | | Learning rate | $0.01, 0.005, 0.001, 0.0001$ | $0.01, 0.005, 0.001, 0.0001$ |
| | | Batch size | $128, 256, 512$ | $256, 512, 1024$ |
| | | LSTM hidden units | $4, 6, \ldots, 30$ | $4, 6, \ldots, 30$ |
| | | LSTM dropout rate | - | - |
| | | BR size | $4, 6, \ldots, 20$ | $4, 6, \ldots, 30$ |
| | | Early Stopping (min delta) | 0.0001 | 0.0001 |
| | | Early Stopping (patience) | 30 | 30 |

Table 19: Hyper-parameters search range for G-Net

| Hyperparameter | Cancer simulation | MIMIC III (SS) |
|---|---|---|
| LSTM layers | 1 | 1 |
| Learning rate | $0.01, 0.005, 0.001, 0.0001$ | $0.01, 0.005, 0.001, 0.0001$ |
| Batch size | $32, 64, 128$ | $32, 64, 128$ |
| LSTM hidden units | $4, 6, \ldots, 30$ | $4, 6, \ldots, 30$ |
| FC hidden units | $4, 6, \ldots, 30$ | $4, 6, \ldots, 30$ |
| LSTM dropout rate | - | - |
| R size | $4, 6, \ldots, 20$ | $4, 6, \ldots, 30$ |
| MC samples | 10 | 10 |
| Early Stopping (min delta) | 0.0001 | 0.0001 |
| Early Stopping (patience) | 30 | 30 |

Table 20: Hyper-parameters search range for Causal Transfomer

| Hyperparameter | Cancer simulation | MIMIC III (SS) |
|---|---|---|
| Transformer blocks | 1 | 1 |
| Learning rate | $0.01, 0.005, 0.001, 0.0001$ | $0.01, 0.005, 0.001, 0.0001$ |
| Batch size | $32, 64, 128$ | $32, 64, 128$ |
| Attention heads | 2 | 2 |
| Transformer units | $4, 6, \ldots, 20$ | $4, 6, \ldots, 20$ |
| LSTM dropout rate | - | - |
| BR size | $4, 6, \ldots, 20$ | $4, 6, \ldots, 20$ |
| FC hidden units | $4, 6, \ldots, 20$ | $4, 6, \ldots, 20$ |
| Sequential dropout rate | $0.1, 0.2, 0.3$ | $0.1, 0.2, 0.3$ |
| Max positional encoding | 15 | 15 |
| Early Stopping (min delta) | 0.0001 | 0.0001 |
| Early Stopping (patience) | 30 | 30 |

Table 21: Hyper-parameters search range for Causal CPC

| Model | Sub-model | Hyperparameter | Cancer simulation | MIMIC III (SS) |
|---|---|---|---|---|
| Causal CPC | Encoder | GRU layers | 1 | 1 |
| | | Learning rate | 0.01, 0.005, 0.001, 0.0001 | 0.01, 0.005, 0.001, 0.0001 |
| | | Batch size | 32, 64, 128 | 64, 128, 256 |
| | | GRU hidden units | 4, 6, . . . , 30 | 4, 6, . . . , 30 |
| | | GRU dropout rate | - | - |
| | | Local features (LF) size | 4, 6, . . . , 20 | 4, 6, . . . , 20 |
| | | Context Representation (CR) size | 4, 6, . . . , 20 | 4, 6, . . . , 20 |
| | | Early Stopping (min delta) | 0.001 | 0.001 |
| | | Early Stopping (patience) | 100 | 100 |
| | Decoder | GRU layers | 1 | 1 |
| | | Learning rate (decoder w/o treatment sub-network) | 0.01, 0.005, 0.001, 0.0001 | 0.01, 0.005, 0.001, 0.0001 |
| | | Learning rate (encoder fine-tuning) | 0.001, 0.0005, 0.0001, 0.00005 | 0.001, 0.0005, 0.0001, 0.00005 |
| | | Learning rate (treatment sub-network) | 0.05, 0.01, 0.005, 0.0001 | 0.05, 0.01, 0.005, 0.0001 |
| | | Batch size | 32, 64, 128 | 32, 64, 128 |
| | | GRU hidden units | CR size | CR size |
| | | GRU dropout rate | - | - |
| | | BR size | CR size | CR size |
| | | GRU layers (Treat Encoder) | 1 | 1 |
| | | GRU hidden units (Treat Encoder) | 6 | 6 |
| | | FC hidden units | 4, 6, . . . , 20 | 4, 6, . . . , 20 |
| | | Random time indices (m) | 10% | 10% |
| | | Early Stopping (min delta) | 0.001 | 0.001 |
| | | Early Stopping (patience) | 50 | 50 |

Table 22: Hyper-parameters search range for Causal CPC

| Model | Sub-model | Hyperparameter | Cancer simulation | MIMIC III (SS) |
|---|---|---|---|---|
| Causal CPC | Encoder | GRU layers | 1 | 1 |
| | | Learning rate | 0.01, 0.005, 0.001, 0.0001 | 0.01, 0.005, 0.001, 0.0001 |
| | | Batch size | 32, 64, 128 | 64, 128, 256 |
| | | GRU hidden units | 4, 6, . . . , 30 | 4, 6, . . . , 30 |
| | | GRU dropout rate | - | - |
| | | Local features (LF) size | 4, 6, . . . , 20 | 4, 6, . . . , 20 |
| | | Context Representation (CR) size | 4, 6, . . . , 20 | 4, 6, . . . , 20 |
| | | Early Stopping (min delta) | 0.001 | 0.001 |
| | | Early Stopping (patience) | 100 | 100 |
| | Decoder | GRU layers | 1 | 1 |
| | | Learning rate (decoder w/o treatment sub-network) | 0.01, 0.005, 0.001, 0.0001 | 0.01, 0.005, 0.001, 0.0001 |
| | | Learning rate (encoder fine-tuning) | 0.001, 0.0005, 0.0001, 0.00005 | 0.001, 0.0005, 0.0001, 0.00005 |
| | | Learning rate (treatment sub-network) | 0.05, 0.01, 0.005, 0.0001 | 0.05, 0.01, 0.005, 0.0001 |
| | | Batch size | 32, 64, 128 | 32, 64, 128 |
| | | GRU hidden units | CR size | CR size |
| | | GRU dropout rate | - | - |
| | | BR size | CR size | CR size |
| | | GRU layers (Treat Encoder) | 1 | 1 |
| | | GRU hidden units (Treat Encoder) | 6 | 6 |
| | | FC hidden units | 4, 6, . . . , 20 | 4, 6, . . . , 20 |
| | | Random time indices (m) | 10% | 10% |
| | | Early Stopping (min delta) | 0.001 | 0.001 |
| | | Early Stopping (patience) | 50 | 50 |

