# OpenReview forum: "Causal Contrastive Learning for Counterfactual Regression Over Time"
_NeurIPS.cc/2024/Conference — NeurIPS 2024 poster_

### Official Review · Reviewer_Nj8s · 2024-07-01

**Soundness:** 3
**Presentation:** 3
**Contribution:** 3
**Rating:** 6
**Confidence:** 3

**Summary:**

the paper proposes a method based on CPC to estimate the causal effect of treatments over a period of time

**Strengths:**

- Interesting paper
- well written , although some things could have been cleaner
- good experimentation and ablation study
- interesting use of cpc

**Weaknesses:**

- Not clear from an intuition perspective if the contrastive approach incentivises the model to learn what the authors want
- The complexity of the method is significant , raising questions about applicability and reproducibility
- The point about invertible processes is not a great one as invertible processes severely limit the function space available without offering much in return. They also increase computational complexity significantly
 - not the first use of a contrastive objective in causal learning , id suggest that this contribution is toned down a bit

**Questions:**

not many, mostly how susceptible is the method to violations of the assumptions the authors have set out ?

**Limitations:**

adequately discussed

---

> ### Author Rebuttal · Authors · 2024-08-07
>
> Thank you very much for your review!
>
> ## Weaknesses
>
> 1. Please see the global rebuttal.
>
> 2. We have reported the complexity of our model and that of baselines for experiments on both synthetic and semi-synthetic data. For instance, in Table 2 of the core paper, we show the training time of our model (aggregated encoder and decoder) and the prediction time for counterfactual trajectories. We provide similar details for MIMIC III data in Appendix F.2.3, Table 11. Although our model has fewer parameters compared to SOTA models, its training time for both the encoder and decoder is comparable to SOTA, even though some baselines, such as the Causal Transformer and G-Net, consist of a single end-to-end trained model.
>
> We employed several techniques to achieve efficiency without sacrificing performance, detailed in Algorithms 1 and 2 for encoder and decoder training.
>
> **Trick 1:** Since computing the InfoNCE loss for all time steps $t = 0, \dots, t_{\text{max}}$ can be computationally intensive, we sample a single time step $t$ per batch, using the corresponding process history $\mathbf{H}\_t$ for the InfoMax objective. The sampled $\mathbf{H}\_t$ is then partitioned into future $\mathbf{H}\_t^f$ and past $\mathbf{H}\_t^h$ sub-processes.
>
> **Trick 2:** The decoder is trained autoregressively and without teacher forcing. For each time step $t$, our GRU-based decoder predicts the future sequence of treatments $\hat{Y}\_{t+1:t+\tau}$ with its hidden state initialized to $\mathbf{\Phi}\_t$. To enhance training efficiency, we randomly select $m$ time indices $t_{i,1}, \dots, t_{i,m}$ for each individual $i$ and compute future treatment response sequences $\hat{Y}\_{i,t_{i,1}+1:t_{i,1}+\tau}, \dots, \hat{Y}\_{i,t_{i,m}+1:t_{i,m}+\tau}$. It is sufficient to train using only 10% of the time steps.
>
>
> 3. The point about invertibility discussed just before Section 5.2 considers the implicit inversion of the encoder. Recent literature on contrastive learning suggests that an encoder-only architecture, when trained with contrastive loss, implicitly approximates an inverse of the true data-generating process. Our argument is that our encoder, trained with the InfoMax loss, might benefit from this byproduct without additional cost. Thus, there is no need for a decoder to reconstruct the input space, potentially reducing computational complexity. To address concerns about invertibility, we have added formal proof to this assertion in response to Reviewer DSyv.
>
>
> 4. In lines 103-114 of our paper, we discuss the novelty of applying the InfoMax principle to causal inference, specifically for time-varying data. Our work extends contrastive learning beyond static settings, as seen in [Chu et al., 2022] and [Zhu et al., 2024]. We provide theoretical arguments showing how selection bias is completely resolved in the representation space, a critical aspect not fully addressed in these static settings. We appreciate your feedback and will consider it carefully to further position our work accurately within the existing research landscape.
>
> ## Questions
>
> 1. The crucial assumption in our work is sequential ignorability, which is necessary for identifying conditional counterfactual responses. This assumption implies no unobserved confounders, whether static or time-varying. We tested our model's robustness by training Causal CPC and baselines on MIMIC III data with some confounders masked. Results are reported in Section 6.4, Table 4 of the core paper. Compared to Table 1, where assumptions are not violated, errors for Causal CPC, Causal Transformer, and CRN increase when confounders are masked, except for RMSN, which remains insensitive during the robustness test. However, RMSN starts to underperform significantly at $\tau \geq 2$. Our model continues to outperform baselines in long-term forecasts even when sequential ignorability is violated, demonstrating its ability to encode long-term dependencies of observed confounders effectively.
>
> Thank you once again for your valuable feedback.
>
> [Chu et al., 2022] Chu, Z., Rathbun, S. L., and Li, S. (2022). Learning infomax and domain-
> independent representations for causal effect inference with real-world data.
>
> [Zhu et al., 2024] Zhu, M., Wu, A., Li, H., Xiong, R., Li, B., Yang, X., Qin, X., Zhen, P., Guo, J.,
> Wu, F., et al. (2024). Contrastive balancing representation learning for heterogeneous dose-response
> curves estimation.

---

> > ### Comment · Reviewer_Nj8s · 2024-08-12
> > **Rebuttal acknowledgement**
> >
> > I acknowledge that I have read the authors rebuttal, and I maintain my score of acceptance

---

> > > ### Author Response · Authors · 2024-08-12
> > >
> > > Thank you for reviewing our rebuttal and for maintaining your positive score. We appreciate your support and consideration.

---

### Official Review · Reviewer_DSyv · 2024-07-07

**Soundness:** 3
**Presentation:** 3
**Contribution:** 3
**Rating:** 6
**Confidence:** 4

**Summary:**

This paper leverages Contrastive Predictive Coding (CPC) with RNN for counterfactual regression over time to provide a compelling alternative to transformer-based approaches (which are challenging to interpret).  By leveraging CPC to capture long-term dependencies and InfoMax for "reconstructable" representation, the method achieves state-of-the-art results in counterfactual estimation.

**Strengths:**

- The method seems sound.

- The paper is well-written.

- Previous work leveraging contrastive learning for causal inference applies only to the static setting with no theoretical grounding. This paper frames the representation balancing problem from an information-theoretic perspective. The authors show that the suggested adversarial game yields theoretically balanced representations using MI's Contrastive Log-ratio Upper Bound (CLUB), computed efficiently.

-  Achieved good empirical performance with fewer patient data.  Good ablation studies.

**Weaknesses:**

- Experiments: performance in random trajectories for the pharmacokinetic-pharmacodynamic model of tumor growth is missing.

- While good results are achieved with fewer patient data, it is important to check if the results are consistent as the patient number increases.

- Line 96: "*...the role of invertible representation in improving counterfactual regression. Here, we introduce an InfoMax regularization term to make our encoder easier to invert.*" The authors provide some intuitions (lines 204-213) in favor of invertibility (beyond reconstructable) but they are not sufficient. A formal proof is needed.

- For a “reconstructable” representation of the process history $H_{t}$,  While a lower bound of InfoMax objective ( regularization) is optimized during pre-training of the encoder, there is no guarantee that finetuning of pre-training will retain the desired “reconstructable” property of the representation.
Hence, the role of invertible (or reconstructable) representation in improving counterfactual regression is not well understood even though ablation studies show improvement when $\mathcal{L}^{Infomax}$ is introduced.

**Questions:**

-  Line 178-179: *$H_{t}$ is a sequence of high-dimensional covariates and the computation of such loss is computationally demanding.*
- Line 165-167: *...where at each horizon the discriminator parameter matrix $\Gamma_{j}$ is of a small number of dimensions since being a map between two lower-dimensional representations*.

Lower bounds are motivated by the problem of high-dimensionality of $H_{t}$.
What is the dimension of $H_{t}, Z_{t}$ and $C_{t}$ in the performed experiments?

**Limitations:**

See Weaknesses.

---

> ### Author Rebuttal · Authors · 2024-08-07
>
> Thank you very much for your review !
>
> ## Weaknesses
>
> 1. We have included the performance evolution on random trajectories for the pharmacokinetic-pharmacodynamic model of tumor growth in Figure 3 of the core paper. This figure illustrates the model's performance across multiple levels of confounding to evaluate its robustness. The precise values of the errors for each confounding level and at each time step are detailed in Appendix E.2.1, Table 6, due to space constraints in the main paper.
>
> 2. In Appendix F.2.2, we have actually increased the number of patients and tested our model in the same setting of train/validation/test as [Melnychuk et al. (2022)]. The Causal CPC still gives better results than SOTA at the majority of the forecasting horizons.
>
>
> 3. Thank you for highlighting this important point. We acknowledge that a formal proof would strengthen the argument for the role of invertibility in improving counterfactual regression. Here, we provide a more formal basis for the claim.
>
> To ensure identifiability in the latent space, we leverage recent advances in causal and disentangled representation learning. Suppose the true data-generating process is given by $\mathbf{H}\_{t} = g(\mathbf{z}\_t)$, where $\mathbf{z}\_t$ represents the true latent factors. In the sequential context, we assume that the same function $g$ generates two historical subsequences:
> $$
> \mathbf{H}\_t^f = g(\mathbf{z}\_t^f), \quad \mathbf{H}\_t^h = g(\mathbf{z}\_t^h)
> $$
> We assume a general dependency of the form:
>
> $$
> p(\mathbf{z}\_t^f \mid \mathbf{z}\_t^h ) = \frac{Q(\mathbf{z}\_t^f)}{Z(\mathbf{z}\_t^h)} \exp(-d(\mathbf{z}\_t^f, \mathbf{z}\_t^h))
> $$
>
> Here, $\mathbf{\Phi}$ is an encoder, and we use the InfoMax regularization term as follows:
>
> $$
> \mathcal{L}^{(InfoMax)}(\mathbf{\Phi}, d, \mathcal{B}) := -\mathbb{E}\_{\mathcal{B}} \left[ \log \frac{\exp(-d(\mathbf{\Phi}(\mathbf{H}\_t^f), \mathbf{\Phi}(\mathbf{H}\_t^h)))}{\sum_{l = 1}^{|\mathcal{B}|} \exp(-d(\mathbf{\Phi}(\mathbf{H}\_{l,t}^f), \mathbf{\Phi}(\mathbf{H}\_t^h)))} \right]
> $$
>
> According to Matthes et al. (2023), under certain conditions, if the encoder $f$ minimizes $\mathcal{L}^{(InfoMax)}$, then $h = g \circ f$ is a scaled permutation matrix. This result suggests that when the encoder achieves a minimizer for $\mathcal{L}^{(InfoMax)}$, the encoder function $f$ closely approximates an invertible transformation of $g$.
>
> From a causal inference perspective, if $Y_{it}(\omega_{it}) \perp W_{it} \mid \mathbf{H}\_{it}$ and $\mathbf{H}\_{it} = g(\mathbf{Z}\_{it})$, then an invertible function $g \circ f$ ensures that:
>
> $$
> Y_{it}(\omega_{it}) \perp W_{it} \mid g \circ f (\mathbf{H}\_{it})
> $$
>
> Thus, $Y_{it}(\omega_{it}) \perp W_{it} \mid g(\mathbf{C}\_{it})$ and since $g$ is invertible, we have:
>
> $$
> Y_{it}(\omega_{it}) \perp W_{it} \mid \mathbf{C}\_{it}
> $$
>
> This demonstrates that the representation $\mathbf{C}\_{it}$ retains the essential independence structure, facilitating accurate counterfactual inference.
>
>
> 4. During the decoder training, the encoder is fine-tuned. However, to ensure that the encoder is only "slowly" optimized during the fine-tuning, we actually choose a learning rate of $5.10^{-4}$, whereas, for the decoder, we ensure a faster convergence by using a ten times higher learning rate of $5.10^{-3}$.
>
> ## Questions
>
> 1. For the cancer simulation data: $\dim(\mathbf{H}\_t) = 4$, $\dim(\mathbf{Z}\_t) = 12$, $\dim(\mathbf{C}\_t) = 14$. For the MIMIC III data: $\dim(\mathbf{H}\_t) = 74$, $\dim(\mathbf{Z}\_t) = 14$, $\dim(\mathbf{C}\_t) = 14$.
>
> Your comment highlights a crucial aspect of our research field. In fact, obtaining suitable datasets to verify counterfactual methods, particularly those applied over time with a considerable number of covariates, is a significant challenge in the causality research field. While benchmark datasets for causal inference in static settings are relatively more abundant, as detailed in the survey by Yao et al. (2021), longitudinal datasets that fit our framework are rare.
>
> Given the increasing interest in causal inference, it is our hope that more longitudinal datasets, especially those with relatively high-dimensional covariates, become publicly available. Such datasets would significantly benefit the research community by providing more opportunities for validation and comparison of different methods.
>
> Despite these challenges, we conducted experiments aligned with baseline methods [Lim et al., 2018; Bica et al., 2020; Melnychuk et al., 2022] using both the Tumor Growth dataset and the challenging MIMIC III dataset. Our results showcased superior performance in long-term counterfactual regression, demonstrating the effectiveness of our proposed method.
>
> Once again, we thank you for your insightful comments and hope our responses address your concerns. We welcome further discussion or questions to clarify any remaining issues.
>
> [Bica et al., 2020] Bica, I., Alaa, A. M., Jordon, J., and van der Schaar, M. (2020). Estimating
> counterfactual treatment outcomes over time through adversarially balanced representations.
>
> [Lim, 2018] Lim, B. (2018). Forecasting treatment responses over time using recurrent marginal struc-
> tural networks.
>
> [Matthes et al., 2023] Matthes, S., Han, Z., and Shen, H. (2023). Towards a unified framework of
> contrastive learning for disentangled representations.
>
> [Melnychuk et al., 2022] Melnychuk, V., Frauen, D., and Feuerriegel, S. (2022). Causal transformer
> for estimating counterfactual outcomes.
>
> [Yao et al., 2021] Yao, L., Chu, Z., Li, S., Li, Y., Gao, J., and Zhang, A. (2021). A survey on causal
> inference.

---

> > ### Comment · Reviewer_DSyv · 2024-08-12
> >
> > I acknowledge that I have read the author's rebuttal. I maintain my score of acceptance.

---

> > > ### Author Response · Authors · 2024-08-13
> > >
> > > Thank you very much for reviewing our rebuttal. We appreciate your consideration and positive assessment of our paper.

---

### Official Review · Reviewer_6PYm · 2024-07-10

**Soundness:** 3
**Presentation:** 3
**Contribution:** 3
**Rating:** 6
**Confidence:** 3

**Summary:**

The paper introduces a novel algorithm for long-term counterfactual forecasting over time through representation learning of historical information and balanced representation learning that predicts the outcome given balanced treatments. The representation learning process combines Contrastive Predictive Coding and Information Maximization, while the training of balanced representation learning involves an adversarial game in the actual outcome and treatment networks.

**Strengths:**

1. The paper introduces a novel algorithm for predicting counterfactual responses over time which outperforms the baselines in both a synthetic and a semi-synthetic dataset.
2. The proposed algorithm is robust.
3. The paper is well-structured and comprehensive, covering theoretical analysis, experiments, ablation studies, and falsifiability tests.
4. The experiment settings are described in detail.
5. Most of the notation is clear and easy to understand.

**Weaknesses:**

1. Is it possible to conduct experiments on real datasets?
2. It appears there is only one synthetic dataset and one semi-synthetic dataset. In the synthetic experiment, is the standard deviation of the errors calculated from experiments with 5 different seeds instead of random settings of the data generating process? Would it be feasible to conduct experiments on multiple synthetic datasets with random settings to obtain error bars, rather than relying on just one synthetic dataset?
3. There don't seem to be any comparisons of time consumption with baselines.
4. The loss functions are constructed based on the lower or upper bound of the mutual information, but there is no discussion or measurement regarding the tightness of the bound.

**Questions:**

1. In equation 1, $U_{t+j}$ in the numerator has one subscript, while $U_{l,t+j}$ in the denominator has two subscripts. Could you please make them consistent? Is $U_{t+j}$ information for one specific individual or an aggregate for all individuals? Is InfoNCE calculated for each individual or as the expectation over all individuals? Given that in equation 1, the subscript is $\mathcal{B}$, is it the individual index
2. Could you elaborate on how Theorem 5.2 influences the construction of the loss function in Equation 7? Specifically, does Theorem 5.2 impact the formulation or design of the loss function described in Equation 7? The authors discuss what we can infer if equality holds from lines 189 to 192, but will equality actually hold? The description in this paragraph suggests that equality can be achieved, yet Proposition 5.1 and Theorem 5.2 indicate positive gaps between the two mutual information measures without specifying the magnitude of these gaps, and the construction of the contrastive loss does not appear to be influenced by Theorem 5.2, as long as Proposition 5.1 holds.
3. Could you explain what is $\gamma$ in equation 7? Is $\gamma$ a similar definition of horizons?
4. In line 202, is there any proof provided regarding "The theoretical existence of such a function"
5. Since the goal is to estimate the counterfactual responses given in the equation at the bottom of page 3, could you explain why we still need to predict the treatments as the conditional expectation is given $ W_{t+1:t+\tau} $? Is this training needed because we want balanced learning? If so, could you explain why selection bias should be avoided given the goal of learning the expectation conditioning on potential treatments?
6. In Theorem 5.3, could you explain why or when the “if” condition holds?
7. Does the balanced learning only work for $W_{t+1}$? Why is it not necessary to extend it to $W_{t+\tau}$ for $\forall \tau\geq 1$?
8. In section 5.3, is $\theta_{R}=[\theta_1, \theta_2]$?

**Limitations:**

The authors have discussed the broader societal impacts.

---

> ### Author Rebuttal · Authors · 2024-08-06
>
> Thank you very much for your feedback!
>
> ## Weaknesses
> 1. and 3. Please see the global rebuttal.
>
> 2. Using multiple synthetic datasets with random settings to obtain error bars is a valuable suggestion, but we face constraints: limited budget for computational resources, increased energy consumption which conflicts with our goal to reduce carbon footprint, and access to only one cluster with a single GPU. Despite these limitations, our methodology remains robust and aligns with recent practices, such as those seen in the Causal Transformer.
>
> 4. Measuring tightness for high-dimensional variables is challenging and computationally intensive, as discussed in [Rainforth et al., 2018; Poole et al., 2019]. This is typically feasible only with low-dimensional toy datasets or Gaussian assumptions. Batch size is crucial, as shown by the inequality in Eq. (3). Larger batch sizes $\mathcal{B}$ tighten the lower bound by increasing $\log(|\mathcal{B}|)$. We used batch sizes of 256 for the encoder and 128 for the decoder to balance feasibility and performance. Despite not having perfectly tight bounds, mutual information and self-supervision provide beneficial inductive bias, as discussed in Appendix C.2. Our ablation study shows that replacing InfoNCE with estimators like NWJ, which has less bias but higher variance, did not improve performance. Our original model, using InfoNCE and InfoMax, outperformed other approaches (see Table 9, Appendix F2.1).
>
> ## Questions
>
> 1. $\mathbf{U}\_{t+j}$ refers to information for a specific individual. To simplify notation, we avoided using a subscript $i$ for individuals and only used double subscripts for batches, as in Eq. (1). We will add this note before discussing our modeling: "To simplify notation, we remove the subscript $i$ from $\mathbf{H}\_{i,t}$ when discussing an individual, unless contextually necessary." Thus, the InfoNCE loss in Eq. (1) is the expectation over all individuals in the batch, denoted by $\mathcal{B}$.
>
> 2. The rationale behind the InfoMax Principle is to maximize the mutual information (MI) between the input process history $\mathbf{H}\_{t}$ and the learned context $\mathbf{C}\_t$. Before Proposition 5.1, we argued that maximizing MI between two representations—one for a historical subsequence $\mathbf{H}\_t^f$ (denoted $\mathbf{C}\_t^h$) and one for a future subsequence $\mathbf{H}\_t^f$ (denoted $\mathbf{C}\_t^f$)—is more beneficial both computationally and from an inductive bias perspective. We needed a theoretical justification for this approach, provided in Proposition 5.1, which shows it represents a lower bound of the original InfoMax term. Theorem 5.2 formally defines the gap between these terms; this gap is positive but can be zero theoretically. When the gap is zero (i.e., equality in Proposition 5.1), Theorem 5.2 indicates this occurs when the learned representations $\mathbf{C}\_t^h$, $\mathbf{C}\_t^f$, and $\mathbf{C}\_t$ satisfy $\mathbf{H}\_t \perp \mathbf{C}\_t^f \mid \mathbf{C}\_t^h$ and $\mathbb{P}\_{\mathbf{C}\_{t}^h \mid \mathbf{H}\_t} = \mathbb{P}\_{\mathbf{C}\_{t}^h \mid \mathbf{C}\_{t}^f}$.
>
> The justification for the loss in Eq. (7) comes not from Theorem 5.2 alone but from its combination with Proposition 5.1, which shows our InfoMax simplification is a valid lower bound. Therefore, the contrastive loss in Eq. (7) is justified since we have $ \log(|\mathcal{B}|) - \mathcal{L}^{(InfoMax)} \leq I(\mathbf{C}\_t^h,\mathbf{C}\_t^f)\leq I(\mathbf{H}\_{t}, (\mathbf{C}\_t^h, \mathbf{C}\_t^f)) $.
>
> 3. In Eq. (7), $\gamma$ refers to the parameters of the non-linear discriminator used in the definition of InfoMax loss.
>
> 4. Yes, indeed! We have included proof in the appendix, specifically Proposition G.2. We will add a reference to it in the core paper.
>
> 5. As you suggested, training is crucial to ensure covariate balance in the representation space. Our goal is counterfactual regression, specifically estimating $\mathbb{E}(Y_{t+\tau}(\omega_{t+1:t+\tau}) \mid \mathbf{H}\_{t+1})$. For simplicity, let $\tau=1$, our goal is to estimate:
> $$
> \mathbb{E}(Y_{t+1}(\omega_{t+1}) \mid \mathbf{H}\_{t+1}) = \mathbb{E}(Y_{t+1} \mid \mathbf{H}\_{t+1}, W_{t+1} = \omega_{t+1}) = f(\mathbf{H}\_{t+1}, W_{t+1})
> $$
> Observed data includes only one treatment regime per individual $i$ at each time step, $W_{i, t+1} = \omega_{i, t+1}$. After fitting the regression model $\hat{f}$, we estimate counterfactual responses for the same individual $i$ under different treatments, $\hat{f}(\mathbf{h}\_{i, t+1}, \omega_{t+1}')$ where $\omega_{t+1}' \neq \omega_{i,t+1}$ (e.g., radiotherapy instead of chemotherapy). The challenge with treatment switching at inference is that $\mathbf{H}\_{t+1}$ and $W_{t+1}$ are not independent. This lack of independence can bias counterfactual estimates, introducing selection bias [Robins, 1999]. To address this, we learned a representation $\mathbf{\Phi}(\mathbf{H}\_{t+1})$ during decoding to remove selection bias.
>
> 6. Theorem 5.3 highlights the importance of optimizing the treatment classifier: maximizing its log-likelihood leads to a smaller divergence $D_{KL}(p(\mathbf{\Phi}\_{t+1}, \omega_{t+1}) \| q_{\theta_W}(\mathbf{\Phi}\_{t+1}, \omega_{t+1}))$. Effective training of the treatment classifier helps meet the “if” condition, validating $I_{\text{CLUB}}$ as an upper bound on MI between representation and treatment. This motivates our adversarial game framework, which alternates between optimizing the treatment classifier and minimizing the CLUB upper bound, as formally proven in Theorem 5.4.
>
> 7. Covariate balancing in the representation space applies beyond $t+1$. For simplicity, we presented it for $t+1$, but in practice, it extends to all forecasting horizons. Our theorem holds for other horizons by replacing $\mathbf{H}\_t$ with $\mathbf{H}\_{t+\tau -1}$ and $W_{t+1}$ with $W_{t+\tau}$.
>
> 8. Yes! We will add it right after Eq. (7).
>
> We hope we have addressed your insightful questions and concerns. Thank you once again.

---

> > ### Comment · Reviewer_6PYm · 2024-08-12
> >
> > Thank you for the detailed answers. My concerns have been addressed. I will maintain the score.

---

> > > ### Author Response · Authors · 2024-08-12
> > >
> > > Thank you for reviewing our responses; we're glad your concerns have been addressed. We appreciate your positive assessment and constructive feedback.

---

### Official Review · Reviewer_PYKS · 2024-07-13

**Soundness:** 3
**Presentation:** 2
**Contribution:** 2
**Rating:** 5
**Confidence:** 3

**Summary:**

This paper presents causal CPC, a framework for predicting counterfactual responses under time-varying treatments. The proposed method consists of two components: an encoder that leverages contrastive predictive coding and infomax principle to learn a representation of the history H, and a decoder that leverages a loss function based on adversarial game to predict outcome and treatment with balanced representations to account for selection bias. The concept of utilizing MI-based principles for casual representation learning in the longitudinal setting is interesting, whose efficiency has been demonstrated by the SOTA. However, I believe the paper could benefit from more streamlined writing with a clearer justification of how the use of MI-based principles enables efficient learning of long-term dependency.

**Strengths:**

1. The concept of utilizing MI-based principles for casual representation learning in the longitudinal setting is interesting.
2. The performance of the model has been validated with a number of datasets, including fully-synthetic and semi-synthetic data.
3. Some learning objectives are backed up with theoretical evidence.

**Weaknesses:**

1. Even thought the paper demonstrates the benefits of using CPC and infomax for learning long-term dependency with RNN, it still remains unclear to me, both in intuition and in methodological details, how this benefit was attained. The paper can benefit from some more insights, particularly in section 5.1, that links the proposed information theoretic methods to their actual benefits. The theoretical analysis seems a bit detached from the context and not so helpful in providing insights for the effectiveness of the proposed method.

2. Following on the second point, in the ablation study, it seems that the removal of the InfoNCE loss is non-significant. This seems somewhat curious to me given the amount of emphasis the authors put on the importance of CPC.

3. Some recent work in counterfactual outcome prediction with time-varying treatments should be discussed in the related work:
 - Berrevoets et al, Disentangled counterfactual recurrent networks for treatment effect inference over time

 - Chen et al, A Multi-Task Gaussian Process Model for Inferring Time-Varying Treatment Effects in Panel Data

 - Wu et al, Counterfactual Generative Models for Time-Varying Treatment

 - Frauen et al, Estimating average causal effects from patient trajectories

**Questions:**

1. How to decide the relative weighting for the encoder and decoder during fine-tuning?
2. Related to Weakness #2, how to better illustrate the role of L_infoNCE given the relatively mild drop in performance after removing it?

**Limitations:**

Limitations were mentioned in the appendix, but not much in the main text. Adding more discussion of model limitations in the Conclusion section might help.

---

> ### Author Rebuttal · Authors · 2024-08-07
>
> Thank you very much for your review!
> ## Weaknesses
>
> **On the importance of the INfoNCE loss**: We have provided extended results pointing out the discrepancy in errors for all forecasting horizons in Table 8, specifically for the semi-synthetic MIMIC-III data. We reported only the average error over all horizons in the core paper due to space constraints.
>
> However, while computing the average error for the Full Model, we made an error by counting the error for the first horizon twice. In fact, from Table 8, **the actual average error of the Full Model is $0.62$ and not $0.66$**. To provide a clear understanding of the importance of regularization terms, we plot the error evolution of models similar to Figure 3 with the actual values of Table 8 **(Figure 4, the one-page pdf)**. The detailed ablation study in Table 8 shows that starting from $\tau =2$, the drop in error when removing InfoNCE loss is consistently at $9\%$.
>
> For the synthetic dataset, we have verified that the reported average error based on the extended table of ablation (Table 7) is correct, and there are no typos. For the synthetic dataset, we should agree that the impact of the InfoNCE loss is still less significant. However, there are reasons related to the nature of the cancer simulation dataset itself, where the dimensionality of the time-varying component of the process history plus static covariates, $\mathbf{U}\_{t}=[\mathbf{V}, \mathbf{X}\_{t}, W_{t-1}, Y_{t-1}]$, is relatively low, with **only four dimensions**. Our model is specifically designed to excel on datasets with a higher number of confounding dimensions, leveraging our modeling and regularizations based on contrastive losses. These findings align with our results on MIMIC-III, where the dimensionality of $\mathbf{U}\_{t}$ is **substantially larger at 72**, and our model consistently outperforms baselines at larger prediction horizons.
>
> **On the suggested recent work in counterfactual regression**: Thank you for your valuable suggestions. We will include the mentioned works in our related work section. However, while these papers provide interesting contributions to the field, we actually did not initially include them for the following reasons that we will gladly to related work discussion:
>
> - **Berrevoets et al. (2021)** focuses on sequences of binary treatments and requires a stronger version of the sequential ignorability assumption: sequential ignorability conditioning on current covariates. However, we assume a weaker version of sequential ignorability, which holds by conditioning on the entire history of the covariates, thus accommodating long-lasting confounding effects.
>
> - **Chen et al. (2023)** considers only binary treatments and targets the estimation of the average effect on the treated. More importantly, the authors assume a specific treatment assignment scenario where pre-treatment and post-treatment regimes can be defined for all treated individuals. In our paper's notation, this implies $W_{it} = 1$ for $g=1$ and $t > T_0$, with $T_0$ being the time of treatment application for all treated individuals and $g$ the index of the two groups (1 for treated, 0 otherwise). Our setting is more general, allowing for a complex assignment mechanism that varies individually and permits arbitrary swings in treatment value. Treatment and control groups vary substantially over time, and we have multiple treatment groups because the treatment is non-binary, making Chen et al. (2023) incompatible with our causal assumptions.
>
> - **Wu et al. (2023)** addresses high-dimensional outcome generation of counterfactuals according to a time-varying treatment plan, adhering to the same sequential ignorability assumption. However, it is not designed for causal forecasting over multiple time steps.
>
> - **Frauen et al. (2023)** is specifically tailored to estimating the average causal effect and cannot be used to estimate conditional counterfactual responses or individual treatment effects, as it targets the marginal expected counterfactual response using the g-computation.
>
> ## Questions
> 1. In fact, there is no relative weighting of the encoder and the decoder during fine-tuning. During pretraining, the encoder is trained using contrastive losses. At the fine-tuning stage, the decoder is trained from scratch using the adversarial game:
>
> $$
> \begin{aligned}
> & \min_{\theta_{R}, \theta_{Y}} \mathcal{L}\_{dec}(\theta_{R}, \theta_{Y}, \theta_{W}) = \mathcal{L}\_Y(\theta_{R}, \theta_{Y}) + I_{\text{CLUB}}(\Phi_{\theta_{R}}(\mathbf{H}\_t), W_{t+1}; q_{\theta_{W}}) \\
> & \min_{\theta_{W}} \mathcal{L}\_{W}(\theta_{W}, \theta_{R}) = -\mathbb{E}\_{\Phi_{\theta_{R}}(\mathbf{H}\_t)} \left[ \log q_{\theta_{W}}(W_{t+1} \mid \Phi_{\theta_{R}}(\mathbf{H}\_t)) \right]
> \end{aligned}
> $$
>
> Instead of freezing the parameters of the encoder, we allow a slow update (small learning rate compared to the decoder) of its parameters following the losses defined in the adversarial game only. During the fine-tuning, there is no reconsideration of the encoder loss made of contrastive terms.
>
> ## Limitations
>
> Thank you very much for this comment. We will provide an updated version of the conclusion by including "**While our model is tailored for long-term prediction, it does not outperform SOTA models for short horizons. However, our design of contrastive loss, especially InfoNCE in Eq. (4), suggests the possibility of deciding on a trade-off between short-term prediction quality and long-term prediction without training the model twice for the two objectives by designing suitable weights for each contrastive term at a given time step. We leave such an investigation for future work**."
>
> We thank you again for your detailed review and valuable feedback, which helped correct a very important typo. If you have any more questions or need further clarification, please feel free to reach out.

---

> > ### Comment · Reviewer_PYKS · 2024-08-12
> >
> > Thanks for the comments. My concerns are addressed and I will raise my score to 5.

---

> > > ### Author Response · Authors · 2024-08-12
> > >
> > > Thank you for reviewing our responses; we’re happy your concerns have been addressed. We appreciate your thoughtful consideration.

---

### Official Review · Reviewer_pP4H · 2024-07-14

**Soundness:** 3
**Presentation:** 3
**Contribution:** 3
**Rating:** 6
**Confidence:** 4

**Summary:**

This paper proposes a new method for counterfactual outcome prediction over time, with the goal of avoiding highly complicated models and expensive computation while achieving state-of-the-art prediction accuracy for time series with long-range dependencies. To this end, the presented method employs RRNs for long-range forecasting, rather than more complicated network architecture like Transformers, together with contrastive predictive coding (CPC) to capture long-range dependencies in the presence of time varying confounders and information maximization (InfoMax) to help with learning invertible representation for addressing the identification issue. Experiments with synthetic and semi-synthetic data are conducted to demonstrate the efficacy and superiority of the proposed method.

**Strengths:**

1. The aim to develop less complicated neural network models to tackle the challenging problem of counterfactual regression over time.
2. The proposed approach is technically sound and the experiment result looks promising overall.
3. The writing is very effective, and the authors have done a great job on clearly presenting the complicated process/components of their method.

**Weaknesses:**

1. The efficacy of the proposed method for long-range predictions is only demonstrated by experiments to certain extend, and from the results, it is not clear how the method would perform when $\tao$ is greater than 10 steps. In particular, from the middle and right diagrams in Fig. 3, the advantage of the  proposed method over the baselines start to drop and some of the baselines start to outperform the proposed method. The paper would be much stronger if some theoretical evidence of the performance of the proposed method could be provided.
2. There are some doubts with the experiment results or performance of the proposed method (please see the Questions section for more details).

**Questions:**

1. Why a mapping of $\textbf{C}_t$, instead of $\textbf{C}_t$ itself is used as input of the decoder part of the proposed architecture?
2. Could you provide any explanation on why the performance advantage of the proposed method starts to drop when $\tao$ becomes bigger (as shown in the middle and right diagrams in Figure 3, and Table 10)? Moreover, why the baselines in general over outperform the proposed method with small horizon predictions?
3. Table 2: the results shown are for the tumor growth data only and $\gamma=1$. How the number of model parameters, training and testing time when different $\gamma$ values are used?
4. Why MSM is not used/reported in all the experiments? Having poor performance may not be a convincing reason for excluding it.

**Limitations:**

The paper has indicated some limitations of the proposed method, e.g. in Fig. 3 and Section 6.4, but more explicit discussions on the limitations in terms of decreased performance/superiority when the prediction horizon is small or large and when there are unobserved confounders should be given.

---

> ### Author Rebuttal · Authors · 2024-08-07
>
> Thank you very much for your review!
>
> ## Weaknesses
>
> We conducted a similar experiment as suggested by the reviewer mdqp, where we reduced the sequence length seen during training but maintained a large forecasting horizon ($\tau$). It is important to note that the middle and right diagrams in Figure 3 refer to a more challenging experiment with a high level of confounding. The main idea behind this experiment is that since the level of confounding is unknown in real data, it is beneficial to test different levels of confounding.
>
> Figures 1, 2, and 3 in the one-page PDF in our global rebuttal show that Causal CPC still outperforms in the new experiment with large forecasting horizons. It is true that in Figure 3, while Causal CPC outperforms in most forecasting horizons, its prediction for the last time step is close to that of CRN. However, this is only because $\tau=10$ is the last forecasting horizon at which the process history was contrasted, and not specifically related to $\tau=10$.
>
> To further support our claims, we reran our model with the initial sequence lengths, and this time with $\tau=15$ and $\gamma=2$. Results in Figure 7 of the one-page PDF show that our model still outperforms baselines at forecasting horizons larger than $\tau=10$ because the encoder is retrained in such a way that the InfoNCE loss is computed across all 15-time steps. We observe that the last prediction error is close to that of the baselines. This suggests that it may be desirable to train the model over larger forecasting horizons than initially intended.
>
>
> ## Questions
>
> *Why is a mapping of $\mathbf{C}_t$, instead of $\mathbf{C}_t$ itself, used as input to the decoder part of the proposed architecture?*
>
> 1. We use a mapping of $\mathbf{C}_t$ to match the decoder's input dimension. This mapping acts as a dimension adapter, allowing the model to be more general. During our experiments, we kept the dimensions of both the mapping and the original representation consistent.
>
> *Could you explain why the performance advantage of the proposed method starts to decline as $\tau$ increases (as shown in the middle and right diagrams in Figure 3 and Table 10)? Additionally, why do the baselines generally outperform the proposed method for short-term predictions?*
>
> 2. The decline in performance with increasing $\tau$ is due to high levels of confounding in the data, which affects all models. This confounding makes it challenging to maintain performance over longer horizons. For short-term predictions, the proposed method is designed for long-term predictions, which may explain why baseline models, optimized for immediate predictions, outperform it in the short term. This decline is observed primarily with the tumor growth dataset, **which has a low dimensionality of covariates (only 4 dimensions)**. However, our model performs consistently on more challenging datasets like MIMIC-III. We will add an experiment where we retrain the models to predict over more than 10 horizons.
>
>
> 3. Due to space constraints, we have included similar results for the MIMIC-III dataset in Appendix F.2.3, Table 11. For the different versions of the cancer simulation according to $\gamma$, there is no substantial increase in the training and prediction times. As we increase the level of confounding, the fine-tuning scheme tends to select a slightly higher dimension for the representation. For example, the dimension of the representation was 14 for $\gamma=1$ and 16 for $\gamma=2$. Overall, the increase in the number of parameters does not exceed $5\%$ in the worst cases across all levels of confounding.
>
> 4. MSM was excluded from Figure 3 to improve readability due to its high error values. However, we made sure to include it in the corresponding detailed table of results (Table 6 Appendix E.2.1). We will especially add MSM results to Table 1, which are the following:
>
> | Model       | $\tau = 1$      | $\tau = 2$      | $\tau = 3$      | $\tau = 4$     | $\tau = 5$      | $\tau = 6$      | $\tau = 7$      | $\tau = 8$      | $\tau = 9$      | $\tau = 10$     |
> |-------------|-----------------|-----------------|-----------------|----------------|-----------------|-----------------|-----------------|-----------------|-----------------|-----------------|
> | MSM Table 1 | 1.20 $\pm$ 0.10 | 1.83 $\pm$ 0.26 | 2.07 $\pm$ 0.40 | 2.3 $\pm$ 0.45 | 2.54 $\pm$ 0.45 | 2.90 $\pm$ 0.37 | 3.01 $\pm$ 0.38 | 3.06 $\pm$ 0.37 | 3.08 $\pm$ 0.36 | 3.09 $\pm$ 0.36 |
>
>
> ## Limitations
>
> As suggested by reviewer PYKS, we will add a discussion of the limitations, particularly regarding the performance in short-term predictions, to the conclusion. Additionally, as you mentioned, we will emphasize this discussion in the falsifiability section. We intend to include at the end of line 358: "We also observe in the falsifiability test, similar to our experiments, that Causal CPC does not particularly outperform baselines in short-term prediction, starting from $\tau=6$. However, in Table 1, when sequential ignorability holds, our model starts to perform similarly to baselines at $\tau=4$ and outperforms them at $\tau=5$. This indicates that the violation of causal assumptions may relatively exacerbate the issue of not outperforming in short-term horizons."
>
> Thank you once again for your feedback and comments.

---

> > ### Comment · Reviewer_pP4H · 2024-08-13
> > **Thanks for your detailed responses**
> >
> > Thanks the authors for your detailed and helpful responses. I will maintain my positive rating.

---

> > > ### Author Response · Authors · 2024-08-13
> > >
> > > Thank you very much for your response; we’re happy our responses were helpful and your concerns addressed.

---

### Official Review · Reviewer_mdqp · 2024-07-14

**Soundness:** 3
**Presentation:** 3
**Contribution:** 2
**Rating:** 6
**Confidence:** 2

**Summary:**

The paper presents a novel approach to counterfactual regression over time, particularly focusing on long-term predictions. It introduces a method that leverages RNNs combined with CPC and InfoMax. This approach aims to capture long-term dependencies in data and improve computational efficiency without relying on complex transformer models.

**Strengths:**

1. Performance: The proposed model achieves superior performance in long-term prediction tasks compared to existing models.
2. Theoretical foundations: The paper provides a thorough theoretical grounding for the proposed method, including a new information-theoretic perspective on representation balancing.

**Weaknesses:**

1. Complexity: While the method avoids transformers, the combination of CPC and InfoMax introduces its own complexity, which may pose challenges for implementation and replication. Also reported in Table 2 and Table 11, where the proposed method requires more training time with fewer parameters compared to the Causal Transformer and G-Net.
2. Scope of evaluation: Although the evaluation is comprehensive, it could be expanded to include more diverse datasets to further validate the robustness and generalizability of the method. For example, 1) what is the total trajectory length in the synthetic experiment? I wonder about the performance and an additional experiment showing the performance under different trajectory lengths would be appreciated. 2) what is the dimensionality of the variables? I wonder about the performance when dealing with the high-dimensional case.

**Questions:**

1. In the problem formulation, the treatment of W is assumed to be discrete, is it possible to extend it to be continuous and would such an extension violate the theoretical result?
2. What is the performance of the approach when dealing with time series under different trajectory lengths and high-dimensional cases?
3. How should the counterfactual treatment be chosen in the experiment? Randomly chosen from the support or do the authors have some rules?
4. Could the authors explain why the proposed method 'excels at large horizon predictions, it does not outperform SOTA models on short-term predictions'?

**Limitations:**

The authors claimed in the Checklist.

---

> ### Author Rebuttal · Authors · 2024-08-07
>
> Thank you very much for your valuable feedback!
> ## Weaknesses
>
> ### Complexity:
>
> It is true that our model requires fewer parameters compared to SOTA models but has a relatively higher training time. However, several points mitigate this concern. The reported training time for CPC includes both encoder and decoder training. Once the encoder is pre-trained, the decoder training takes under 10 minutes, which is faster than the Causal Transformer and CRN. From a practical perspective, prediction time is also crucial. When estimating counterfactual responses, multiple trajectories per individual are generated, significantly increasing the dataset size. The Causal Transformer is designed to be trained using teacher forcing, which is disabled during testing, requiring multiple updates and batch reloads to perform counterfactual regression over multiple horizons. G-Net faces a similar issue, relying on time-consuming Monte Carlo sampling. However, our model is substantially faster than baselines at prediction time and outperforms them on large horizons, offering a good trade-off between computational efficiency and prediction quality.
>
> ### Scope of evaluation and question 2:
>
> In our cancer simulation data, the maximum sequence length is $60$, with a projection horizon of $10$, as noted in the paper. For the MIMIC III data, sequence lengths range from a minimum of $20$ to a maximum of $100$, varying due to common censorship phenomena in real applications. During training, a flag variable indicates active entries per individual, ensuring that the contrastive loss is computed only when the time step is "active." The dimensionality of the variables was $4$ for the synthetic experiment and $74$ for the MIMIC semi-synthetic experiment.
>
> We recognize the importance of additional experiments with varying sequence lengths. To further challenge our model, we reduced the sequence length to $40$ for synthetic data while keeping the forecasting horizon at $\tau = 10$. We also decreased the maximum sequence length for semi-synthetic data from $100$ to $60$. As shown in Figure 1,2,3 for cancer simulation and Figure 5 for MIMIC III of the one-page PDF, the error evolution mirrors the core paper's results. Despite these challenges, our model continues to outperform most baselines in long-term forecasting.
>
> However, it is important to note that our model does not outperform others in short-term forecasting, a limitation observed even in additional experiments. Nevertheless, its strong performance in long-term predictions underscores its potential for applications where long-term accuracy is crucial. These additional experiments further validate the robustness and generalizability of our method across varying trajectory lengths.
>
> **Questions:**
>
> 1. From a theoretical perspective, it is possible to extend the theory to handle continuous treatments by replacing the treatment classifier with a treatment regressor. Since we maximize the likelihood, there will be no change in the definition of equilibrium in Theorem 5.4. However, the results should be reported as $\mathbf{H}\_t \perp W_{t+1}$ instead of conditioning on each treatment value. The significant changes would affect the proof of Lemma G.3, where we find the best treatment classifier given a fixed representation $q(W_{t+1} = 1 \mid \Phi(\mathbf{H}\_t)), \dots, q(W_{t+1} = K \mid \Phi(\mathbf{H}\_t))$. For continuous treatments, we would need to find an infinite number of values $q(W_{t+1} = \omega \mid \Phi(\mathbf{H}\_t))$. Rigorously, the standard optimization problem in Eq. (18) should be converted to a functional optimization problem, where the goal is to find the best continuous density function $q$. Under mild regularity conditions, variational calculus can be applied to the continuous version. Thus, Lemma G.3 can be extended, and the rest of the proof should follow easily by converting summations to integrals. We would be glad to add a formal extension to Theorem 5.4 in the appendix.
>
> From a practical perspective, continuous treatments would be represented as a single dimension in training, whereas discrete treatments are represented by a $K$-dimensional vector (one-hot encoding). This might pose a risk that the treatment information is not adequately captured during counterfactual predictions when it is continuous. One possible solution is to discretize the continuous treatments, which would make the problem more straightforward for our model.
>
>
> 3. We detailed the generation rules in the Experimental Protocol (Appendix D). For the cancer simulation data, we generated counterfactual treatments similarly to CRN and Causal Transformer papers. Trajectories are generated with a single treatment per trajectory, while the treatment slides over the forecasting range to generate multiple trajectories. The idea is to choose the best time step to apply chemo or radiotherapy and not apply any treatment in subsequent steps to see how tumor volume will evolve. For MIMIC semi-synthetic data, trajectories are generated such that at each time step, treatment is randomly chosen from the support. Therefore, treatments can be applied at multiple time steps for one counterfactual sequence of treatments.
>
> 4. Please see the global rebuttal.
>
> Thank you for your insightful comments and suggestions.

---

> > ### Author Response · Authors · 2024-08-10
> > **Correction of a typo in rebuttal**
> >
> > In response to Question 1, line 2:  "However, the results should be reported as $\Phi(\mathbf{H\_t}) \perp W\_{t+1}$ ...".

---

> > ### Comment · Reviewer_mdqp · 2024-08-12
> >
> > Thank the authors for their clarification and now I have a better understanding of the details. Although I still have some concerns about the higher training time, it is acceptable to me now and I will raise my score. Also it could be better if the authors could polish the figures, as the current font size is somewhat small and difficult to read clearly.

---

> > > ### Author Response · Authors · 2024-08-12
> > >
> > > Thank you very much for your thoughtful feedback and for raising the score. We appreciate your suggestion about the figures and will work specifically on improving them for clarity in the final version.

---

### Author Rebuttal · Authors · 2024-08-07

## Global Rebuttal

We would like to first thank the reviewers for their very constructive feedback and valuable comments.

1. We can use real data, like the MIMIC III dataset, instead of its semi-synthetic version. However, evaluating counterfactual trajectories is not possible due to the absence of counterfactual responses in real datasets. Upon the request of reviewer 6PYm, we assessed our model and baselines by forecasting factual responses over time and estimating responses for each individual's observed treatment trajectory. Our experiments on real MIMIC III data, shown in Figure 6 of the 1-page PDF, demonstrate that our model consistently outperforms all baselines at large horizons, highlighting the effectiveness of our model design.

2. To further consolidate the discussed intuition in lines 204-213 regarding the implicit inversion of the encoder, going beyond the reconstructability of the representation classically advocated by the InfoMax principle, we provided a proof sketch upon the request of reviewer DSyv in favor of this claim.

3. Following the suggestion of reviewer mdqp, we added additional experiments where we reduced the sequence length from 60 to 40 for synthetic data while keeping the forecasting horizon at $\tau = 10$. We also decreased the maximum sequence length for semi-synthetic data from 100 to 60. As shown in Figures 1,2,3, and 5 of the one-page PDF, the error evolution mirrors the core paper's results. Our model continues to outperform most baselines in long-term forecasting.

5. We would like to further emphasize the intuitions behind the design of the InfoNCE loss, as raised by reviewers Nj8s and PYKS, which can also explain why our model excels at large prediction horizons but not in the short term, a question raised by reviewers pP4H and mdqb. Our main intuition is to learn a representation of $\mathbf{H}\_{t+1}$ that is highly predictive of the future components of the process, i.e., $\mathbf{U}\_{t+j}=[\mathbf{V}, \mathbf{X}\_{t+j}, W_{t+j-1}, Y_{t+j-1}]$ for $j=1, \dots, \tau$. To achieve this, we learn a representation of the process history leveraging an RNN, referred to as the context $\mathbf{C}\_t$, and aim to make it predictive, at large horizons, of different future local representations $\mathbf{Z}\_t$ of process components $\mathbf{U}\_{t+j}$. We ensure such predictiveness using the InfoNCE loss $\mathcal{L}^{(InfoNCE)}\_j$, for all the prediction time steps $j=1, \dots, \tau$.

**To encourage the context $\mathbf{C}\_t$ to learn the shared information between all future local representations, particularly future covariates**, we minimize the InfoNCE loss averaged across all future time predictions via $\mathcal{L}^{CPC}:= \frac{1}{\tau} \sum_{j = 1}^{\tau} \mathcal{L}^{(InfoNCE)}\_j$. We can also trivially write as given in Eq. (5):

$$
\frac{1}{\tau} \sum_{j = 1}^{\tau} I(\mathbf{U}\_{t+j}, \mathbf{C}\_{t}) \geq \log(|\mathcal{B}|) - \mathcal{L}^{CPC}.
$$

Therefore, as we minimize $\mathcal{L}^{CPC}$, **we push the model to learn a context representation that shares the maximum information with the future components across all prediction time steps.** This approach encourages the model to capture the **global structure of the process (when $\tau$ is large)**, which is very beneficial for performing counterfactual regression over large horizons. It is, therefore, expected, given the tailored design for long-term prediction, that the model does not necessarily outperform SOTA models at short-term horizons. We further demonstrated the importance of the InfoNCE loss using an ablation study.

6. Finally, we compared the time consumption of our model with baselines in Table 2 of the core paper for a synthetic dataset with a confounding level ($\gamma = 1$), including both training and prediction times. The same table for the semi-synthetic dataset is in Table 11 of Appendix F.

Thank you for your insightful comments and suggestions.

---

### Decision · Program_Chairs · 2024-09-25

**Decision:**

Accept (poster)

**Comment:**

The paper presents a novel approach to counterfactual regression over time, i.e., long-term predictions. It combines RNNs with CPC and InfoMax. This approach aims to capture long-term dependencies in data without relying on complex transformer models. There is a general   consensus among the reviewers that the paper contains interesting ideas and will make non-negligible contributions to the field. The reviewers raised several important concerns, most of which were addressed in the authors' rebuttal. It should be possible for the authors to address the remaining concerns in the camera-ready version of the manuscript.

Hence, I recommend that the paper be accepted as a poster.